# How Deep Can the Endophytic Mycobiome Go? A Case Study on Six Woody Species from the Brazilian Cerrado

**DOI:** 10.3390/jof9050508

**Published:** 2023-04-25

**Authors:** Jefferson Brendon Almeida dos Reis, Georgios Joannis Pappas Junior, Adriana Sturion Lorenzi, Danilo Batista Pinho, Alexandra Martins Costa, Mercedes Maria da Cunha Bustamante, Helson Mario Martins do Vale

**Affiliations:** 1University of Brasília (UnB), Institute of Biological Sciences, Department of Phytopathology, Brasília 70910-900, DF, Brazil; 2University of Brasília (UnB), Institute of Biological Sciences, Department of Cellular Biology, Brasília 70910-900, DF, Brazil; 3University of Brasília (UnB), Institute of Biological Sciences, Department of Ecology, Brasília 70910-900, DF, Brazil

**Keywords:** *Ascomycota*, tropical savannah, *Diaporthe*, ITS, metabarcoding, mycobiota, *Dothideomycetes*, *Sordariomycetes*

## Abstract

Elucidating the complex relationship between plants and endophytic fungi is very important in order to understand the maintenance of biodiversity, equity, stability, and ecosystem functioning. However, knowledge about the diversity of endophytic fungi from species of the native Brazilian Cerrado biome is poorly documented and remains largely unknown. These gaps led us to characterize the diversity of Cerrado endophytic foliar fungi associated with six woody species (*Caryocar brasiliense*, *Dalbergia miscolobium*, *Leptolobium dasycarpum*, *Qualea parviflora*, *Ouratea hexasperma*, and *Styrax ferrugineus*). Additionally, we investigated the influence of host plant identities on the structure of fungal communities. Culture-dependent methods coupled with DNA metabarcoding were employed. Irrespective of the approach, the phylum *Ascomycota* and the classes *Dothideomycetes* and *Sordariomycetes* were dominant. Using the cultivation-dependent method, 114 isolates were recovered from all the host species and classified into more than 20 genera and 50 species. Over 50 of the isolates belonged to the genus *Diaporthe*, and were distributed into more than 20 species. Metabarcoding revealed the phyla *Chytridiomycota*, *Glomeromycota*, *Monoblepharomycota*, *Mortierellomycota*, *Olpidiomycota*, *Rozellomycota*, and *Zoopagomycota*. These groups are reported for the first time as components of the endophytic mycobiome of Cerrado plant species. In total, 400 genera were found in all host species. A unique leaf endophytic mycobiome was identified in each host species, which differed not only by the distribution of fungal species, but also by the abundance of shared species. These findings highlight the importance of the Brazilian Cerrado as a reservoir of microbial species, and emphasize how endophytic fungal communities are diversified and adapted.

## 1. Introduction

Fungi constitute one of the most diverse groups of eukaryotic organisms [1,2,3], and play numerous ecological roles [4,5,6]. Among several fungal lifestyles, endophytic behavior stands out due to a series of benefits to the host plants [7,8,9,10,11]. Unlike the soil dwelling saprophytic species, endophytic fungi reside inside plant tissues, growing within roots, trunks and leaves without causing an apparent disease [12,13,14,15]. These microorganisms evolved along with their host plants, establishing a mutualistic ecological system where both benefit from the interaction [11]. In many cases, endophytic fungi depend on the host plant to complete part or all of their life cycle [16,17,18] and produce appressoria to allow the infection of the respective hosts [11,19]. They usually require protection and nutrients from host plants, and in return fungi contribute to the host’s growth and nutrient uptake.

Because of the co-evolution process, endophytic fungi display variable degrees of specialization depending on the host plant range they are able to colonize [13,14,15,16,17,18]. The host plant identity strongly predicts the pool of fungal species capable of endophytic colonization [18,19,20,21]. Therefore, the endophytic mycobiome is pre-defined by a set of factors including the degree of fungal specialization in colonizing one or more host plants and the host identity factor (genotype and phenotype), which acts as an environmental filter. The effect of the host plant identity and the degree of fungal specialization have already been reported in plant species from the most varied terrestrial ecosystems [15,20,22,23]. However, in plant species from the Cerrado biome (Brazilian Neotropical savannah), the composition of the endophytic mycobiome and the effect of the host plant identity on the fungal community is still neglected and poorly studied.

Cerrado is the second-largest biome in Brazil and one of the most biodiverse ecosystems in the world [24,25], with a unique phytophysionomic composition [26] that harbors over 10,000 species of vascular plants [27,28]. The savanna-like Cerrado vegetation in turn represents an important ecological niche for microbial species [29,30], especially for endophytic fungi [30,31]. However, the endophytic mycobiome from Cerrado plant species and the effect of the host plant identity on its taxonomic composition have been sparsely explored over the years, using culture-dependent approaches [31]. Although extremely useful [14,23,32], these methods lack the resolution to access the real diversity of endophytic fungi. This occurs because of the presence of non-culturable taxa, variations in growth rates of isolates that makes their separation and recovery difficult, chances of contamination with saprophytic fungi from the environment [33,34,35], and the fact that most endophyte studies only use the ITS region as a genetic marker, which is not recommended [36]. On the other hand, DNA metabarcoding has proven to be a powerful tool for inferring ecological data and assessing the endophytic fungal community [15,34,35,37], since it covers a high diversity of organisms in large scale studies, including uncultivable fungi by conventional culture-dependent methods and artificial conditions [37]. Therefore, the present study aims to characterize the leaf endophytic mycobiome of six Cerrado woody species, namely, *Caryocar brasiliense* Camb. (*Caryocaraceae*), *Dalbergia miscolobium* Benth. (*Fabaceae*), *Leptolobium dasycarpum* Vogel Yakovlev (*Fabaceae*), *Qualea parviflora* Mart. (*Vochysiaceae*), *Ouratea hexasperma* (A.St.-Hil.) Baill. (*Ochnaceae*), and *Styrax ferrugineus* NeesandMart. (*Styracaceae*) using culture-dependent and culture-independent methods, and to evaluate the effect of the host identity on the richness and abundance of endophytic fungal species from the mycobiome of the host plants.

## 2. Materials and Methods

### 2.1. The Study Area

The study was conducted in an area of cerrado *sensu stricto* located in the *Reserva Ecológica do Instituto Brasileiro de Geografia e Estatística* (RECOR/IBGE) (15°55’ S, 47°51’ W) (Figure 1a), which is 35 km south of Brasília, DF, km 0 of BR−251, Brazil. In this region, the climate is tropical (Aw) with two distinct seasons: the dry season, from April to September, and the rainy season from October to March [38]. The average annual precipitation for the region is between 1100 and 1600 mm [39]. The vegetation of the studied area is characterized by the presence of grasses together with the tree and shrub strata, and woody cover ranging from 10% to 60% [40]. Trees are generally low and crooked, with irregular and twisted branches [41] (Figure 1b). The plots where the leaves of native Cerrado plants were collected are part of the project “*Estudo dos Efeitos das Mudanças Globais que Determinam a Estrutura e o Funcionamento dos Ecossistemas do Cerrado*” (IAIPRONEX), implemented at RECOR/IBGE in 1998.

### 2.2. Sampling and Collection

Plant material was collected at the end of the rainy season on 25, 26 and 30^th^ March 2021. Six species of woody plants present in the control plots of the experimental area (Figure 1a), namely, *Caryocar brasiliense*, *Dalbergia miscolobium*, *Leptolobium dasycarpum*, *Qualea parviflora*, *Ouratea hexasperma*, and *Styrax ferrugineus*, were selected (Figure 2). Three individuals per species were sampled, and 20 healthy leaves without apparent symptoms of the disease were collected per individual, totaling 60 leaves per species. The collected samples were stored in sterile plastic bags, packed in thermos boxes with controlled temperatures, and transported to the laboratory where they were washed with neutral soap and running water [42]. After washing, the leaves were subjected to a superficial disinfection process and processed within 24 h.

### 2.3. Surface Disinfection and Sample Processing

From the 60 leaves collected per woody species, 30 were submitted to the superficial disinfection process as used by [15,42], with some modifications. For this, the leaves were submerged in Erlenmeyer flasks containing ethanol (70%) for two minutes, followed by immersion in NaClO (4%) for five minutes and, finally, five successive washes with autoclaved distilled water were carried out. The water from the last rinse was collected in a 15 mL Falcon tube and half the volume was seeded on Potato Dextrose Agar (PDA) medium in duplicate; the other half was subjected to DNA extraction for the validation of the disinfection method.

After surface disinfection, the samples were dried in a flow chamber and processed to be analyzed by dependent and independent methods of cultivation. For this, the protocols by [43,44] were used with adaptations: ten leaves per sampled individual were overlapped and, with the aid of eyelet pliers, they were perforated 15 times, generating approximately 150 leaf fragments with a diameter of five millimeters. Then, eight leaf fragments for each sampled individual (24 fragments per species) were deposited in Petri plates containing PDA (four fragments per plate) supplemented with 250 mg L^−1^ streptomycin and 100 mg L^−1^ chloramphenicol, and incubated at 25 °C in the dark for up to five days. The remaining fragments were placed in sterile 50 mL Falcon tubes and stored in a freezer at −20 °C for further eDNA extraction (environmental DNA) to carry out independent culture analyses by DNA metabarcoding.

### 2.4. Isolation, Purification and Deposit of Cultivable Fungi

As mycelial fungal growth was evident from the edge of the plant tissue arranged in the PDA medium, the fungus was reisolated by collecting a mycelial fragment from the edge of the colony in evidence with sterile needles and transferring it into a new plate containing PDA for further purification. This step was repeated whenever necessary until a monoculture of an endophytic fungus strain was reached with a uniform colony. Purification was carried out by transferring the fungi to Water Agar (WA), followed by incubation at 25 °C for three days. After the incubation period, with the aid of a magnifying glass (Leica M205C^®^) and sterile needle, small fragments of the culture medium containing an individual hypha were collected, deposited in PDA medium, and incubated at 25 °C for five days for subsequent deposit in the Culture Collection at the University of Brasília (CCUB) and DNA extraction. Pure fungal colonies were stored in cryotubes according to the CCUB deposit specifications.

### 2.5. DNA Extraction and Molecular Identification of Isolates

Genomic DNA (gDNA) was extracted from pure cultures using the Wizard^®^ Genomic DNA Purification Kit. For this, approximately 400 mg of mycelium were scraped from the surface of the PDA plate and transferred to 2 mL microtubes containing 700 µL of Nuclear Lysis Solution, 40 µL of TE (Tris-EDTA, pH 8.0), 100 mg of polyvinylpyrrolidone and four steel beads. A negative control was included, containing all components except mycelium. The microtubes were placed in a macerator at 4000 rpm for 90 s, followed by incubation in a dry bath at 65 °C and 500 rpm for 30 min. After incubation, 200 µL of Protein Precipitation Solution was added to each microtube, followed by vigorous shaking for 30 s. The mixture was centrifuged at 14,000 rpm for ten minutes. After centrifugation, 700 µL of the supernatant was recovered and transferred to new 1.5 mL microtubes containing 700 µL of ice-cold propanol. The microtubes were placed in a −20 °C freezer for at least one hour and then centrifuged at 14,000 rpm for ten minutes for DNA precipitation. The supernatant was discarded, and the sediment present in each microtube was washed twice with 700 µL of 70% ethanol and dried under ambient conditions. The pellet was rehydrated with 40 µL of TE (Tris-EDTA, pH 8). Extraction success and gDNA quality were evaluated on 1% (*w*/*v*) agarose gels stained with GelRedTM and visualized under ultraviolet light after electrophoresis in 1× TBE running buffer at 80 V for 30 min.

The 18S-ITS1-5.8S-ITS2 region was PCR amplified from the extracted gDNA using the V9G (Forward 5′–TTACGTCCCTGCCCTTTGTA–3′) [45] and LR5 (Reverse 5′–TCCTGAGGGAAACTTCG–3′) [46] primer pair. Amplification was performed in a 15 µL volume containing 1X PCR Buffer (200 mM Tris-HCl, 500 mM KCl), 0.2 mM deoxyribonucleotide pool, 1.5 mM MgCl_2_, 1 mM primer pair, 1.0 U Taq DNA Polymerase (Invitrogen, Carlsbad, CA, USA), 0.2 mM gDNA and ultrapure deionized water to the predefined final volume. PCR reactions were cycled as follows: five minutes of initial denaturation at 95 °C, followed by 35 cycles of 30 s of denaturation at 95 °C, 45 s of primer annealing at 55 °C, one-minute extension at 72 °C, and a final extension of five minutes at 72 °C. A negative control (no gDNA) was included to validate the PCR amplification. The presence of PCR products (fragments > 800 bp) was analyzed on 1.0% agarose gels after electrophoresis as described previously using the 1 kb molecular marker.

In addition, PCR products were purified using the ExoSAP-IT™ PCR Product Cleanup Reagent (Thermo Fisher Scientific, Waltham, MA, USA) prior to sequencing for the removal of primers and nucleotides. Purified amplicons were sequenced using the same pair of primers previously employed for PCR amplification. The generated nucleotide sequences were treated for noise removal using the DNA Sequence Assembler v4 [47] (https://www.dnabaser.com/download/DNA-Baser-sequence-assembler/index.htmll, accessed on 7 May 2021). Taxonomic assignments of the sequences were made by pairwise comparisons with other sequences deposited in the GenBank of the National Center for Biotechnology Information (NCBI) (http://www.ncbi.nlm.nih.gov/, accessed on 8 May 2021) using the Basic Local Alignment Search Tool (BLAST) to verify sequence identities. Sequence identities (%) lower than 97% were classified only at the genus level. The nucleotide sequences generated in this study were deposited in the GenBank under the accession numbers OP922124–OP922237.

### 2.6. Metabarcoding Sequencing

#### 2.6.1. eDNA Extraction

For eDNA extraction and purification, different protocols were adopted with various adaptations [22,48]. Initially, samples previously frozen (−20 °C) to avoid the oxidation of plant material were vigorously macerated in liquid nitrogen with a mortar and pestle until the formation of a slightly whitish powder. Approximately 400 mg of the macerate was transferred to 2 mL microtubes containing 700 µL of CTAB extraction buffer (7–10% CTAB, 100 mM tri-HCl, 20 mM EDTA–pH 8.0, 1.4 mM NaCl), 150 mg of polyvinylpyrrolidone 40 and 2−5 µL β-mercaptoethanol. This step was performed on a 2 mL microtubes cooling rack. Two negative controls were included, one containing all the required components except the macerated to validate the sterility of the buffer solution, and the other containing all the components except the macerated plus autoclaved ultrapure water (Milli-Q) collected from the mortar and pestle washing to validate their sterility conditions.

The microtubes holding the mixture were incubated in a dry bath at 65 °C for 30 min, and vigorously agitated every five minutes. After incubation, 700 µL of chloroform: isoamyl alcohol (24:1) was added to each microtube and mixed per smooth reversal for ten minutes. The mixture was centrifuged at 14,000 rpm for ten minutes in a non-refrigerated centrifuge, and the aqueous phase was transferred to new 1.5 mL microtubes containing 55 µL CTAB (7%). An additional 700 µL of chloroform: isoamyl alcohol (24:1) was added to each microtube, which were all agitated by smooth reversal for ten minutes and centrifuged under the same previous conditions. This step was repeated until no interface between the aqueous and organic phases was found.

Approximately 700 µL of the aqueous phase was recovered and transferred to new 1.5 mL microtubes containing 700 µL of cold propanol and 30 µL of KOAc (5M), which were agitated by smooth reversal for five minutes and taken to the freezer at −20 °C for a period of two hours. Subsequently, the microtubes were centrifuged at 14,000 rpm for DNA precipitation. The supernatant was discarded and the pellet present in the microtube was then washed, once with 700 µL of 70% ethanol and up to five times with 700 µL of ethanol PA (99.8%). Between each wash, microtubes were centrifuged at 14,000 rpm for two minutes. The pellet was dried at room temperature for one hour and rehydrated using 100 µL of TE (tris-EDTA, pH 8). The integrity of the extracted eDNA was evaluated on 1% (*w*/*v*) agarose gels stained with GelRed^®^ and viewed under ultraviolet light after electrophoresis, as described previously.

#### 2.6.2. Library Construction and Sequencing

The extracted eDNA was used for library preparation and sequencing. The target sequence was the ITS1 genetic marker accessed by the primer pair ITS1-1F-F (5′–CTTGGTCATGGGAAGTAA–3′) and ITS1-1F-R (5′–GCTGCGTCTCTCGTGC–3′). The sequencing was performed with the Miseq 25,000 platform (Illumina, San Diego, CA, USA). Raw sequences were filtered with a quality score ≥ 29 (a precision of ≥99.87%). Paired-end reads were merged into unique readings using Qiime 1.9.0 [49] to produce consensus sequences and increase the annotation accuracy. Adapters were removed using the Qiime script “*split_library.py*” [49], and sequences grouped using Uclisto V1.2.22q [50]. Chimeric sequences were removed with Uchime [51]. Operation Taxonomic Units (OTU) assignment was performed with a 0.97 confidence limit against the Unite ITS1 database with the UNITE 7.2 (UNITE+INSD Dataset) database using the Assign Taxonomy method [52].

### 2.7. Foliar Analysis

In the laboratory, six leaves per individual were scanned (Epson Perfection V700 photo, 600 dpi) and dried in an oven at 40 °C until constant weight. After drying, these same leaves were weighed to calculate specific leaf area (SLA) and leaf area using ImageJ software.

The analysis of the foliar nutritional composition of the six sampled plant species was carried out at the Laboratory of Analysis of Soils, Plant Tissues, and Fertilizers of the Federal University of Viçosa (Universidad Federal de Viçosa, Minas Gerais, Brazil). For this, 10 leaves per individual of the same species were selected, totaling 30 leaves per plant species. The leaves were oven-dried at 40 °C until constant weight and underwent a perchloric nitric digestion process and evaluation of phosphorus (P), potassium (K), calcium (Ca), magnesium (Mg), aluminum (Al), and sulfur (S) contents, followed by the ascorbic acid colorimetric method for P, flame photometry for K, turbidimetric method for S and atomic absorption spectroscopy for Ca, Mg, and Al. Nitrogen (N) contents were evaluated by quantification with the Kjeldahl method. 

### 2.8. Statistical Analyses

Statistical analyses were performed using R language (version 4.1.3) scripts and the QIIME2 workflow [53]. The annotation of the taxonomic sharing network was conducted in the CYTOSCAPE software version 3.9. 1 [54].

Alpha and beta diversity metrics were calculated using the “*vegan*”, “*betapart*” and “*BiodiversityR*” R packages [55,56,57]. The strategies used to assess alpha diversity were the Hill series (q), Pielou evenness, and Faith’s Phylogenetic Diversity [55,56,57,58]. The beta diversity (similarities and dissimilarities) between communities was calculated using several methods, namely, the Jaccard distance (Binary matrix of absence and presence), Bray–Curtis, and the Sorensen beta diversity [56,58]. Relative abundance and abundance curves were measured using the “*dplyr*” and “*tidyverse*” R language packages.

For eDNA metabarcoding analysis, data were previously tested for normality (Shapiro–Wilk test). As the OTUs richness data from endophytic fungi did not satisfy the normality criteria and homogeneity of variance, then the effect of the host plant identity on fungal diversity estimates was examined by the non-parametric Kruskal–Wallis (KW) test. Principal coordinate analysis (PCoA) was used to visualize differences in the endophytic fungal community composition based on the Bray–Curtis dissimilarity. Multivariate permutational analysis of variance (PERMANOVA) was applied to test possible variations in the endophytic fungal community among host species [55]. Differential species abundance analyses were evaluated using the QIIME2 ANCOM45 plugin, consecutively testing each taxonomic level (from phylum to species) to detect significant changes in endophytic fungal community abundances among the different host species [59]. The ANCOM [60] and Metastats [61] tests were applied to find differential abundances of OTUs among the six host plant species.

Rarefaction and extrapolation curves were calculated and plotted to estimate the sampling coverage of metabarcoding data using the R “*iNEXT*” package (iNterpolation/EXTrapolation) with lower and upper 95% confidence limits [62]. Principal component analysis (PCA) was carried out using the R language function “*prcomp*”. 

Network analysis was performed to identify co-occurrence patterns of OTUs of endophytic fungi at different taxonomic levels among the six host plant species sampled. For this, only abundant OTUs were considered (>1000 reads) [20,63]. Shared networks were visualized using CYTOSCAPE version 3.9.1 [53]. The structural attributes of the networks were explored using the “*NetworkAnalyzer*” tool [54].

## 3. Results

### 3.1. Analysis of the Nutritional Composition of Leaves

Leaf nutrient content, leaf area, and SLA varied among the six host species (Table 1). The main differences among the six species were highlighted by Principal Component Analysis (PCA) (Figure 3). *C. brasiliense* was correlated with a high leaf area (cm^2^) and lower leaf nitrogen concentration when compared to other species. *D. miscolobium* demonstrated the opposite profile compared to *C. brasiliense*, with a lower leaf area (cm^2^) and a higher concentration of leaf nitrogen, and higher SLA. *L. dasycarpum* and *Q. parviflora* were associated with lower leaf S, N, Mg, and K contents and higher leaf Al concentration. *O. hexasperma* and *S. ferrugineus* were correlated with higher Mg concentrations, a higher leaf area (cm^2^), and lower SLA and N contents.

### 3.2. Cultivation-Dependent Method

#### 3.2.1. Molecular Identification of Isolates

A total of 114 endophytic fungi were isolated with varying numbers among the host species: *Q. parviflora* (10), *C. brasiliense* (11), *O. hexasperma* (16), *L. dasycarpum* (19), *D. miscolobium* (29), and *S. ferrugineus* (29). Based on the 18S-ITS1-5.8S-ITS2 region of rDNA, the isolates were classified into 48 species (Table 2). However, this number may be even higher because many isolates were assigned up to the genus level.

All isolates belonged to the phylum *Ascomycota* (100%) and were grouped into three major classes: *Sordariomycetes* (80%), followed by *Dothideomycetes* (19.2%) and *Leotiomycetes* (0.8%), seven orders, 14 families and 22 genera. The order *Diaporthales* and the family *Diaporthaceae* were dominant in all host species (Figure 4a,b). *Diaporthe* was the most abundant and ubiquitous genus in all host plants, with 71 taxa (62% of the total isolates) reported, and was distributed in over 20 species (Figure 4c,e). In the host plant *C. brasiliense*, 91% of the endophytic fungi isolated belong to the genus *Diaporthe*, followed by 89.5% in *L. dasycarpum*. *Diaporthe* spp. corresponded to >15% of the total number of isolates in *Q. parviflora*, *O. hexasperma*, *D. miscolobium*, and *S. ferrugineus*. The other 21 fungal genera were found in the different hosts with low abundance. *Didymella* and *Colletotrichum* were the second-most-abundant genera found, each comprising 4.3% of the total isolates. The other genera isolated corresponded to <2.5%. The most diverse host species at the genus level of endophytic fungi were *D. miscolobium*, *O. hexasperma*, and *S. ferrugineus* (Figure 4d).

A heterogeneous mosaic of the number of isolates and species was displayed for each host species. The host plant with the highest number of species of endophytic fungi was *D. miscolobium* (18 species), followed by *S. ferrugineus* (14 species), *L. dasycarpum* (12 species), *O. hexasperma* (11 species), and *Q. parviflora* (8 species) (Figure 5). The host with the lowest number of species of endophytic fungi found was *C. brasiliense*, with only five species. However, more than 50% of the *C. brasiliense* isolates were classified only at the genus level.

#### 3.2.2. Co-occurrence Networks of Endophytic Fungi from the Cultivation-Dependent Method

Most of the endophytic fungal genera isolated were found in only one host plant species (Figure 6a). The unique common genus found in all host species was *Diaporthe*. Other genera (*Coniochaeta*, *Cytospora*, *Dydimella*, *Epicoccum*, *Kalmusia*, and *Neopestalotiopsis*) were isolated from more than one host plant species. At the species level, the majority of the isolates were exclusive for a specific host plant (Figure 6b). Although *Diaporthe* was the common genus displayed among the six host plants, not all identified species of this genus were found in more than one host. The species *D. actinidiae*, *D. schini*, and *D. rosae* were exclusive to the host *C. brasiliense*; *D. cynaroidis*, *D. macadamiae* and *D. rosiphthora* were exclusive to *L. dasycarpum*; *D. parapterocarpi* and *D. raonikayaporum* were exclusive to *D. miscolobium*; *D. baccae* and *D. ilicicola* were exclusive to *S. ferrugineus*, and *D. maytenicola* to the host *O. hexasperma*.

#### 3.2.3. Alpha and Beta Diversity

The community of cultivable endophytic fungi associated with the six species of host plants indicated variations among species diversity per host, as judged by alpha diversity metrics (Figure 7). The cultivable fungal community associated with *D. miscolobium* presented the highest values of alpha diversity for five out of six metrics shown by the Hill series (q), with indexes of species richness (S’) (q = 0), Shannon–Wiener (H’) (q = 1), Simpson’s dominance index (1/D) (q = 2), Gini–Simpson index (q = 3) and Tsallis index (HCDT) (q = 4). The cultivable fungal community present in *O. hexasperma* showed the highest equity (q = 5, Renyi index = 1.8161076). The lowest values of alpha diversity presented by the Hill series were observed in the host *C. brasiliense*.

Based on the species composition of the cultivable fungal communities, it was possible to discriminate four major clusters according to the Bray–Curtis and Jaccard distance (Figure 8a). The fungal community from *C. brasiliense* was quite similar to the community of *D. miscolobium*, and the *L. dasycarpum* community to that of *S. ferrugineus*. The fungal communities isolated from *O. hexasperma* and *Q. parviflora* were positioned in separated clades, with the cultivable endophytic fungi community present in *Q. parviflora* showing the highest dissimilarity values when compared to the others. To better visualize the community structure of cultivable endophytic fungi from the six host species, we used Principal Coordinate Analysis (PCoA) based on the Bray–Curtis distance matrix (Figure 8b). With 37.34% of the variance explained, it was possible to observe that the fungal communities from *Q. parviflora* and *O. hexasperma* were more dispersed than the communities isolated from the other host plants.

The pairwise comparison generated by the Sorensen beta diversity index points out the main components responsible for the dissimilarities found from the community of cultivable endophytic fungi between host plants (Figure 9). Species exchange (turnover) was the main factor responsible for the dissimilarities between hosts. The highest values of turnover were found for the fungal community of *S. ferrugineus* when compared to the isolated communities of *D. miscolobium* (turnover = 0.07692308) and *Q. parviflora* (turnover = 0.07526882). To a lesser extent, the highest value of species loss (nestedness) was observed from the isolated community of *S. ferrugineus* compared to *C. brasiliense* (nestedness = 0.04255319). The Sorensen total value (turnover + nestedness) indicated that the fungal community of *Q. parviflora* was the one that presented the greatest losses and replacements in relation to the isolated communities of *D. miscolobium*, *L. dasycarpum*, and *O. hexasperma*.

### 3.3. Metabarcoding Analysis

#### 3.3.1. Taxonomic Attributions

Of the 18 samples sent for sequencing (three samples per plant species), 16 generated data by metabarcoding sequencing, producing 1,877,129 short-reads after the filtering process. A total of 3821 operational taxonomic units (OTUs) were resolved based on sequence similarity (≥97%). The host *C. brasiliense* had the highest number of OTUs (n = 2050), followed by *O. hexasperma* (n = 1113) and *L. dasycarpum* (n = 859). The taxonomic grouping of OTUs recovery of 442 taxa was: 9 assigned to the phylum level, 35 to the class, 86 to the order, 214 to the family, and 435 to the genus (Table 3). For *Q. parviflora* and *C. brasiliense*, more than 50% of the readings were classified only at the Fungi kingdom level. The largest number of taxa was found in *C. brasiliense* (n = 309 genera), followed by *O. hexasperma* (n = 243 genera), and *L. dasycarpum* (n = 169 genera). *D. miscolobium* and *Q. parviflora* showed the lowest numbers of taxa classified at the genus level.

The rarefaction and extrapolation analysis suggested that the number of OTUs from the foliar endophytic fungal community of the six host species displayed different magnitude values for the Hill numbers (q = 0, q = 1, and q = 2), reaching saturation with the sampling employed (Figure 10). Diversity analysis predicted that the overall OTUs richness (q = 0) ranged from 173 to 2050 (Figure 10a). The indicator q = 1 showed that the number of OTUs equally common ranged from 4 to 165, with values greater than 50 observed only for *C. brasiliense* (q = 1, n = 165) and *O. hexasperma* (q = 1, n = 50) (Figure 10b). For the estimator of dominant species (q = 2), the highest values of equally dominant OTUs were found for *C. brasiliense* (q = 2, n = 28), followed by *L. dasycarpum* (q = 2, n = 13.1) (Figure 10c). The lowest values for q = 1 and q = 2 were observed for *S. ferrugineus*.

#### 3.3.2. Relative Abundance

Nine phyla were found in the six host species, namely *Ascomycota*, *Basidiomycota*, *Chytridiomycota*, Glomeromycota, Monoblepharomycota, Mortierellomycota, *Olpidiomycota*, *Rozellomycota*, and *Zoopagomycota*. *Ascomycota* was the most abundant phylum in all host plant species (Figure 11a). In *S. ferrugineus*, 97% of the reads obtained belonged to *Ascomycota*, followed by 90% in *D. miscolobium*, and 80% in *L. dasycarpum*. Interestingly, more than 60% of the taxonomic attributions made for reads belonging to *C. brasiliense* and *Q. parviflora* were classified only at the Fungi kingdom level. The host *O. hexasperma* showed the highest abundance values for the phylum *Basidiomycota* (15%). For the other host species, *Basidiomycota* represented <2% of the relative abundance. Other phyla accounted for less than 1% of reads. 

The *Dothideomycetes* class was dominant in all host species, followed by *Sordariomycetes*. At the order level, the distribution of taxa varied among host plants (Figure 11b). In *S. ferrugineus*, *Asterinales* was the most abundant order (average 73%), followed by *Hypocreales* (average 9.6%) and *Capnodiales* (8.3%). For *Q. parviflora* and *C. brasiliense*, *Capnodiales* was the most abundant in both plant species, with mean abundances of 31% and 24%, respectively. In *O. hexasperma*, *Pleosporales* (average 39.0%), *Diaporthales* (average 12.0%), and *Tremelales* (average 9.7%) were the most abundant orders. *Pleosporales* (44.0%) and *Diaporthales* were the most abundant orders in *D. miscolobium*, and *Capnodiales* (average 37.0%) and *Diaporthales* (average 27.0%) in *L. dasycarpum*.

The relative abundance at the family level varied among the six host plant species (Figure 12a,b). *Parmulariaceae* was the most abundant family in *S. ferrugineus* (75%), while *Mycospharellaceae* was the most abundant in *Q. parvilhora (30%)* and *C. brasiliense* (16%) (Figure 12b). *Diaporthaceae* (30%) comprised the most abundant family in *L. dasyrcarpum*, Didymosphaeriaceae (23%) in *O. hexasperma*, and *Didymellaceae* in *D. miscolobium*.

At the genus level, each host species presented varying levels of abundance for different fungal genera (Appendix A). The most abundant OTUs at the genus level for *C. brasiliense*, *L. dasycarpum*, *O. hexasperma*, *Q. parviflora*, and *S. ferrugineus* are shown in Figure 13. For the host species *D. miscolobium*, the 25 most abundant OTUs are shown in the Appendix A. Briefly, the genus *Parmularia* was rather abundant in *S. ferrugineus* (75%), *Paramycosphaerella* in *C. brasiliense* (13%), *Madagascaromyces* in *Q. parviflora*, and *Kalmusia* in *O. hexasperma* (Appendix A). In *D. miscolobium*, *Didymella* and *Paramycospharella* were the most abundant genera, and in *L. dasyrcarpum* the most abundant were *Diaporthe* and *Paramycospharella*.

PCA based on the five most abundant OTUs of each host species showed that the first axis (Dim1) explained 18.6% of the variations in the abundance of each OTU, while the second axis (Dim2) described 15.2% of the fluctuation, totaling 33.8% (Figure 14). The host communities were clustered in distinct four-squares. Both axes contributed positively to the abundance of OTUs for *C. brasiliense* and *L. dasycarpum*. The second axis contributed to the variations of the most abundant OTUs in *Q. parviflora* and *S. ferrugineus*.

#### 3.3.3. Alpha and Beta Diversity

The influence of host species identity on richness, diversity, and the evenness of the leaf endophytic fungal community from six host plants was evaluated by different alpha diversity indices. The fungal community found in *C. brasiliense* presented the highest values in all ordinations shown by the Hill Series (q), including the indices that measure species richness (q = 0), diversity (q = 1), and those that give more weight for the most abundant species (q = 2, 3, 4, and 5) (Figure 15a). The second-highest values for the ordinations that measure species richness and diversity (q = 0, 1, respectively) were observed in *O. hexasperma*. The second-highest values for the indices that give more weight to dominant species were observed in *L. dasycarpum*. The lowest values for orders q = 1, 2, 3, 4, and 5 were observed in *S. ferrugineus*. Statistical differences shown by the Kruskal–Wallis Pairwise test (*p* < 0.05) were found by applying the Pielou evenness index, an index that measures the distribution of individuals between species, and in the phylogenetic diversity of Faith, an index that measures the species diversity. Species in a given niche based on the species number present and the phylogenetic differences among them are shown in Figure 15b,c. 

The low Pielou evenness values coupled with the ordination shown by the Hill series (q = 1, 2, 3, 4, 5) for the host plants *S. ferrugineus* and *Q. parviflora* suggested that there was low equitability in the distribution of the endophytic fungi species occurring in these hosts, probably due to a dominant taxonomic group. On the other hand, evenness values close to 1.0 for the hosts *C. brasiliense*, *D. miscolobium*, *L. dasycarpum*, and *O. hexasperma* strongly suggested that the distribution of individuals (reads) among the endophytic fungal species was more equitable. The high PD index value found for the endophytic fungal community from *O. hexasperma* shows that the fungal species were not phylogenetically close, which may suggest endophytic colonization by different evolutionary lines.

Principal coordinate analysis (PCoA) derived from the Bray–Curtis distance showed a pattern of uniform distribution between biological replicates from the same host species and differences in grouping by species that did not overlap (Figure 16a). A total of 39.8% of the variations found among the host species were explained by the X axis (22.6%) and Y axis (17.2%). When the endophytic mycobiota of different host plant species were compared by the Kruskal–Wallis paired test, no significant differences were observed (*p* > 0.05). Although no significant differences exist, this does not necessarily mean that the endophytic fungal communities were homogeneous in terms of taxonomic composition. The pairwise comparison generated by Sorensen’s beta diversity index (presence/absence matrix) pointed out the main components responsible for the dissimilarities among the foliar endophyte communities from the six host plants (Figure 16b). The exchange of species (turnover) was the main factor responsible for the differences among host species.

#### 3.3.4. Differential Abundance

Differential abundance analyses were performed to identify which fungal groups were responsible for the variation between communities of the different host species. The ANCOM test showed considerable statistical differences in the abundance percentage of the phyla *Mortierellomycota* (W = 7) and *Chytridiomycota* (W = 7) for the host species *C. brasiliense* compared to the other host plants analyzed (Figure 17a). The phylum *Basidiomycota* (W= 5) presented the highest percentage of abundance for the host species *O. hexasperma*. The classes *Ustilaginomycetes* (W = 32), *Malasseziomycetes* (W = 30), and *Pezizomycetes* (W = 31) were also more abundant in *O. hexasperma* (Figure 17b). The Metastats test pointed out the main genera that may be contributing significantly to differences among communities of endophytic fungi from the analyzed species of host plants (Figure 17c).

#### 3.3.5. Co-occurrence Networks of Endophytic Fungi for Metabarcoding

The co-occurrence network at the genus level presented the highest number of nodes (number of genera = 99) compared to the other taxonomic levels (Figure 18). Numerically, the node numbers for order and family were 65 and 81, respectively (Figure 18a−c). The number of edges for order was 88, for family 99, and for genus 130. For all the analyzed taxonomic levels, the diameter and radius of the network were 1. The density of the network showed variation (order = 0.021; family = 0.017; gender = 0.013). From the three taxonomic levels considered (order, family, and genus), only one element was present simultaneously in the six species of host plants, namely *Capnodiales* (order), *Mycosphaerellaceae* (family), and *Paramycosphaerella* (genus). Other elements were shared from two to five host plants (Appendix A).

## 4. Discussion

### 4.1. Cultivable Endophytic Fungi

Our findings showed that the community of cultivable endophytic fungi of woody plant species from the Brazilian Cerrado was rather diverse in terms of genera and species, corroborating previous studies [31,64,65] and reinforcing further the importance of this biome as a reservoir of microbial species. To the best of our knowledge, this is the first report of a leaf endophytic fungi community associated with the host species *L. dasycarpum* and *Q. parviflora*, and also one of the first to gather information on the diversity of cultivable endophytic fungi from six woody plant species from the Cerrado biome simultaneously.

In our study, the cultivable endophytic fungal isolates belonged to the phylum *Ascomycota*, mainly to the classes *Sordariomycetes* and *Dothideomycetes*. Similar findings have already been reported in plant species from other ecosystems [14,66,67]. In these studies, although ascomycetes make up the dominant group of the cultivable endophytic fungal community, basidiomycetes occurred in a smaller proportion. The absence of fungal isolates belonging to the phylum *Basidiomycota* in our study may be related to our sampling effort, which was probably not sufficient to cover rare species (Appendix A). In addition, basidiomycetous endophytes are more often found in woody tissues than in foliage [68]. Other aspects may also be considered to explain the absence of basidiomycetes, including a significantly greater number of genes associated with nutrition, sugar metabolism, stress tolerance capacity, and competitive abilities in *Ascomycota* to adapt to different conditions and become a dominant group in specific ecological niches, despite the *Basidiomycota*’s larger genome [69]. 

Using the cultivation-dependent method, 114 isolates were recovered from all the woody host plant species. Although intrinsic limitations to delimiting species may arise using the ITS region [36,70,71] as a unique genetic marker, this region allowed us to classify all the isolates at the genus level, reaching species delimitation in some cases. This was enough to indicate community structure differences in terms of diversity/occurrence of genera and/or species from the different host plant species analyzed. Although the ITS region had not provided species resolution for all isolates, it was useful for establishing similarities (dissimilarities) in the pool of cultivable endophytic fungi from the analyzed host species. In addition, the ITS region provided the sufficient taxonomic resolution to guide further studies and solve close taxa using other genetic markers [71,72].

Based on the ITS sequencing, the 114 isolates were classified into 50 species belonging to 22 genera. Our findings are close to results found by [31] on the diversity of leaf endophytic fungi associated with plant species from the Cerrado and Pantanal biomes. Among our isolates, the genus *Diaporthe* was dominant in the fungal community for all analyzed hosts, as previously shown in surveys of Cerrado plants [31,73]. This genus comprises over 1,100 species, classified as plant pathogens, non-pathogenic endophytes, or saprobes, and capable of producing a range of bioactive secondary metabolites with antimicrobial, antitumor, and trypanocidal properties, among others [73]. Other genera found in our study (*Alternaria*, *Phyllosticta*, *Epicoccum*, *Colletotrichum*, *Fusarium*, *Stenocarpella*, and *Lasiodiplodia*) considered fast-growing [11] have already been reported as endophytic fungi that occur in Cerrado species [31,64,65], in addition to other ecosystems [14,67]. However, as far as we know, the genera *Ascochyta*, *Clathrosporium*, *Dendrothyrium*, *Hymenopleella*, *Kalmusia*, *Melanconis*, *Monochaetia*, *Seiridium*, and *Stilbospora* are reported for the first time as endophytic fungi associated with native Cerrado plant species. Among these genera, *Clathrosporium* and *Hymenopleella* were never reported as endophytic fungi. These findings reveal an unexplored occurrence and diversity of endophytic fungi associated with Cerrado plant species.

Most isolated fungal genera/species were found in only one host plant species. This is highlighted by the genera/species sharing network between host plants (Figure 6). The variability of fungal species isolated from different host plant species in the same habitat was reported in previous studies of cultivable endophytic fungi [23,31,74,75] and leaf litter fungi [76]. One explanation for these variations refers to the “environmental filtering” effect exerted by the host plant identity in conjunction with the plant–fungus symbiont coevolution relationship [20,21,77]. In this context, the phytochemistry and nutritional resources from plant tissues may act as selection factors for endophytic plant colonization [20,21,77]. Moreover, the genotype and phenotype of host plants may drastically influence the occurrence, diversity, and abundance of species within the endophytic fungi mycobiome. Although our sampling effort was not sufficient to cover the full diversity of cultivable endophytic fungi, a larger sample size could have resulted in fewer fungi appearing to be host-specific, as reported by [76].

### 4.2. Characterization of The Endophytic Mycobiome by eDNA Metabarcoding

Culture-independent methods such as DNA metabarcoding have been widely used to study communities of endophytic fungi in plant species from diverse terrestrial ecosystems [20,34,35,78]. However, this is one of the first studies employing metabarcoding to characterize the foliar endophytic fungal community of Cerrado plant species, especially *C. brasiliense*, *D. miscolobium*, *L. dasyrcarpum*, *O. hexasperma*, *Q. parviflora*, and *S. ferrugineus*. In this way, our data unveil information on the richness, abundance, and structure of the endophytic mycobiome from typical Cerrado plant species.

The high biodiversity of the plant endophytic mycobiome from other ecosystems [20,34,79] was corroborated in our study. Metabarcoding resulted in 3821 fungal OTUs distributed heterogeneously among the six host species analyzed. Generally, a high number of OTUs was observed in studies on soil fungal diversity [80,81], considered one of the most biodiverse ecological niches in the world [82].

In our non-cultivated samples, the phylum *Ascomycota* was the dominant endophytic fungal community from all host species analyzed, followed by *Basidiomycota* in a smaller proportion. In terms of taxonomic composition, this feature is commonly reported from communities of endophytic fungi [20,34,37]. Similarly, as reported previously for cultivation-dependent methods, the predominance of ascomycetes may be associated with the metabolic versatility and adaptive plasticity found within the numerous species belonging to this phylum [69]. Additionally, metabarcoding data suggested the presence of the phyla *Monoblepharomycota*, *Mortierellomycota*, *Olpidiomycota*, *Glomeromycota*, *Chytridiomycota*, *Rozellomycota*, and *Zoopagomycota*, which were recently proposed by [83]. To the best of our knowledge, this is the first report of these phyla as members of the phyllosphere from Cerrado plant species. In particular, the presence of *Glomeromycota* in aerial parts of the host species raises important questions on the evolution of symbiotic fungi [78]. *Glomeromycota* comprise a well-known group of fungi strictly associated with roots [84]. Recently, [78] reported for the first time on the phylum Glomeromycota as member of the endofesral mycobiome of aerial parts of *Vaccinium myrtillus*. 

We also reported the occurrence of 35 fungal classes, with the prevalence of *Dothideomycetes* and *Sordariomycetes*, which is also verified in other studies [20,34,37]. Furthermore, our metabarcoding data revealed the presence of more than 400 fungal genera, which could be even higher taking into consideration that most of OTUs have not reached a taxonomic resolution below family. Additionally, this high richness at the genus level represents mostly endophytic fungi, since the eDNA extraction from the last rinse water did not show the presence of DNA when analyzed on 1% agarose gels stained with gelred. However, PCR amplification was not performed on the extracted eDNA targeting the ITS region. We assumed that the high richness found at the genus level predominantly represented endophytic fungi, but it may partially account for epiphytic fungi because the surface disinfection method does not completely eliminate the DNA from leaf surfaces [85]. Nevertheless, the results generated here suggest that the foliar endophytic fungal community of native Cerrado plant species is highly diverse and rather underestimated by cultivation-dependent methodologies. Moreover, we also found genera belonging to *Ascomycota* and *Basidiomycota* (*Russula*, *Cortinarius*, *Geopora*, *Gyroporus*, *Sebacina*, *Tricholoma*, and *Zasmidium*) with an ectomycorrhizal trophic style according to the FungalTraits [86]. These findings reinforce the need for large-scale studies to extend knowledge on evolutionary mechanisms, lifestyle, and the occurrence of fungi in the phyllosphere of the Cerrado biome.

Interestingly, each host plant species harbored a unique mycobiome, with heterogeneous occurrence, richness, and dominance of orders, families, and genera below the class level. *Asterinales/Parmulariaceae/Parmularia*, *Pleosporales/Didymellaceae/Dydimella*, *Capnodiales/Mycospharellaceae/Paramycosphaerella*, and *Diaporthales/Diaporthaceae/Diaporthe* were the main dominant taxonomic groups among the host plant species analyzed. These groups are known to establish numerous symbiotic and parasitic ecological relationships with plant species [20,34,87,88,89,90,91].

As previously stated, most of the OTUs could be classified into higher taxonomic levels (kingdom, phylum, class, order, and family), particularly for the hosts *C. brasiliense* and *Q. parviflora*, where more than 50% of the reads obtained were not fully assigned. This particular finding raises some important questions on how unexplored the Cerrado endophytic mycobiome is. Nevertheless, methodological limitations should not be disregarded since unresolved OTUs may belong to taxonomic groups described morphologically, but without sequences available in reference databases, which makes it impossible to compare and establish the taxonomic delimitation for the identification of novel taxa [35,71]. Considering these points raised, metabarcoding as a large-scale method for the taxonomic identification of complex environmental samples via the analysis of DNA sequences for short regions of one or a few genes, called DNA barcodes, is providing important new insights into the ‘hidden diversity’ of fungi [35,92,93].

Although the analysis of the alpha diversity of the fungal communities indicated no significant differences (*p* < 0.05) in the Hill series, the fungal community present in *C. brasiliense* showed the highest values for all ordinations of Hill’s series. *C. brasiliense* had a larger leaf area (cm^2^) compared to the other host species (Figure 15) and this feature favored more contact opportunities between leaves and aerosols containing fungal spores. Significant differences between the fungal communities from the six host species were found in the Pielou evenness and PD index. These variations in alpha diversity values are expected, since each host plant tissue comprises a unique ecological niche with a gradient of environmental variations and resource availability that influences the taxonomic structure of the endophytic mycobiome [20,21,77].

In terms of differences in the diversity of species from one environment to another [20,94], we used beta diversity indices to compare fungal communities between host species. Although samples from the same host species (i.e., replicas) were grouped together based on their similarity (Figure 16) and did not overlap with communities from other hosts, significant differences were not observed. In terms of composition (occurrence and abundance), the fungal community found in *Q. parviflora* was the most dissimilar compared to communities present in other host species. *Q. parviflora* is an aluminum accumulator (Table 1 and Figure 3). The high concentrations of aluminum in the leaf tissue of this species may explain this community inequality. In general, community dissimilarities are associated with a loss of species from one host to another. As discussed previously for abundance levels and alpha diversity metrics, these dissimilarities may be associated with the ecological niche effect that plant identity and host tissues exert on the fungal community.

In terms of the communities shared among the plant hosts, it was observed in the co-occurrence network that only *Capnodiales/Mycospharellaceae*/*Paramycosphaerella* were common taxonomic groups among the six host plant species. *Capnodiales* are the second largest order in *Dothideomycetes* and encompass ecologically diverse fungi [91]. The trophic styles of this group include saprobes, plants and animal pathogens, mycoparasites, lichens, and epi- and endophytes [91]. This high versatility in the trophic style of this group may explain the ubiquitous occurrence of *Paramycosphaerella* among the six host species. The lack of shared groups highlights the importance of the host identity factor in the composition of endophytic fungal communities among Cerrado plants, since the host species may exert an environmental filtering effect [20,21,77]. In this sense, the genotype and phenotype of the host plant may be selecting the pool of species that will colonize it endophytically.

## 5. Conclusions

Our study extends the knowledge on the diversity of endophytic fungi from the Cerrado biome, and reveals a highly rich habitat containing unusual fungal representatives with an ectomycorrhizal trophic style that has never been reported previously as components of the endophytic mycobiome of aerial parts of Cerrado plant species. In terms of occurrence, richness, and abundance of species, each host species harbored a unique endophytic mycobiome. With the exception of *D. miscolobium*, and considering only OTUs classified at the genus level, over 400 fungal genera were observed in the six host plant species analyzed. This reinforces the importance of the Cerrado as a reservoir of species, especially fungal species, and points to conservation efforts of this threatened biome. In addition, considerable differences in taxonomic composition were observed, employing cultivation-dependent and -independent methods, with DNA metabarcoding showing the diversity and richness of fungal species not observed by the cultivation-dependent method. Despite the promising results produced by metabarcoding, it is strongly encouraged to use a combination of methods, since cultivation-independent techniques measure ecological data and cultivation-dependent approaches provide fungal isolates for further bioprospection studies. Isolates of unexplored endophytic fungi from the Brazilian Cerrado are particularly important, since they might produce novel bioactive compounds with biological activities for medical and/or biotechnological applications with unique characteristics.

## Figures and Tables

**Figure 1 jof-09-00508-f001:**
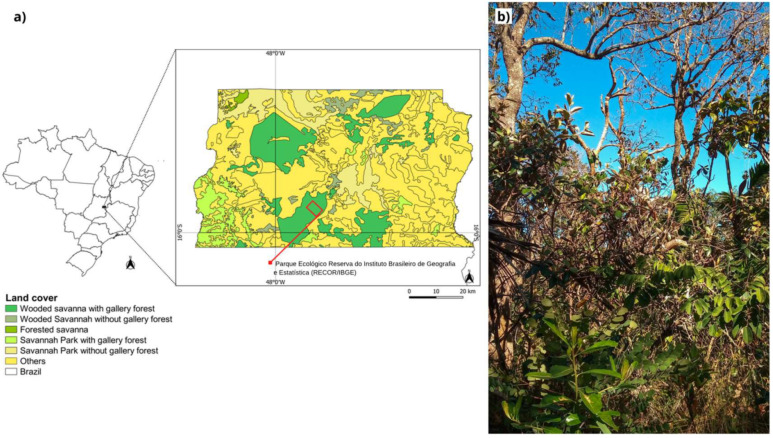
Sampling area. (**a**) Map of the Federal District of Brazil and land cover; (**b**) vegetation of the experimental area. Figure 1a was generated using QGIS Version 3.26.2 software.

**Figure 2 jof-09-00508-f002:**
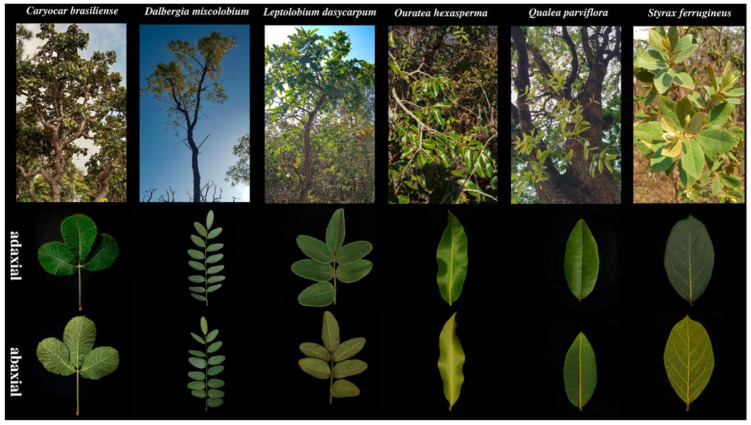
Plant species analyzed and their respective leaf morphologies.

**Figure 3 jof-09-00508-f003:**
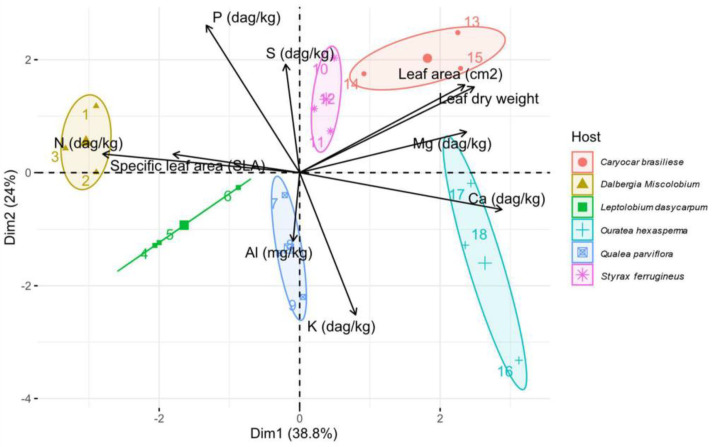
Principal Component Analysis (PCA) for detecting the correlations among the variables of leaf nutrient concentration, specific leaf area (SLA–cm^2^/g) and leaf area (cm^2^) from the six host plant species. Dim 1 (PC1) and Dim 2 (PC2) represent the x and y axes, respectively. The ellipses show biological replicas per host species. SLA: Specific Leaf Area; Al: Aluminum; Ca: Calcium; K: Potassium; Mg: Magnesium; P: Phosphorus; S: Sulfur. Figure created by R software version 4.1.1. using the package “*ggplot2*”.

**Figure 4 jof-09-00508-f004:**
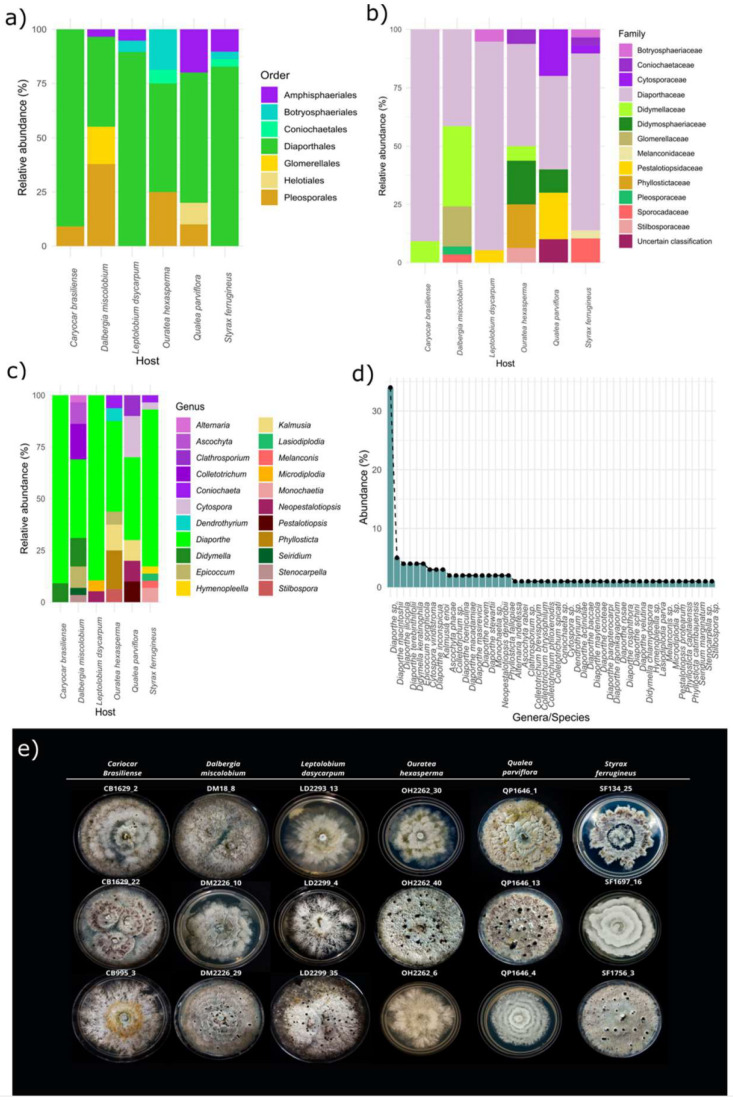
Relative abundance at different taxonomic levels of cultivable endophytic fungi from six host species of the Brazilian Cerrado. (**a**) Relative abundance at order level, (**b**) family, and (**c**) genus. In (**d**), the relative abundance of the total isolates at genus and species level is shown. In (**e**), isolates of endophytic fungi belonging to the genus *Diaporthe*, and cultivated in PDA at 27 °C without lighting for 15 days are shown. The codes CB1629-2, CB995-3, DM18-8, DM2226-10, DM2226-29, LD2299-4, OH2262-30, OH2262-40, OH2262-6, QP1646-13, and SF1697-16 correspond to *Diaporthe* species unidentified. CB1629-22: *Diaporthe schini*; LD2293-13: *Diaporthe stewartii*; LD2299-35: *Diaporthe macadamiae*; QP1646-1: *Diaporthe macintoshii*; QP1646-4: *Diaporthe inconspicua*; SF134-25: *Diaporthe foeniculina*; SF1756-3: *Diaporthe alicycle*. Figures (**a**–**d**) were created with R software version 4.1.1. using the package “ggplot2”.

**Figure 5 jof-09-00508-f005:**
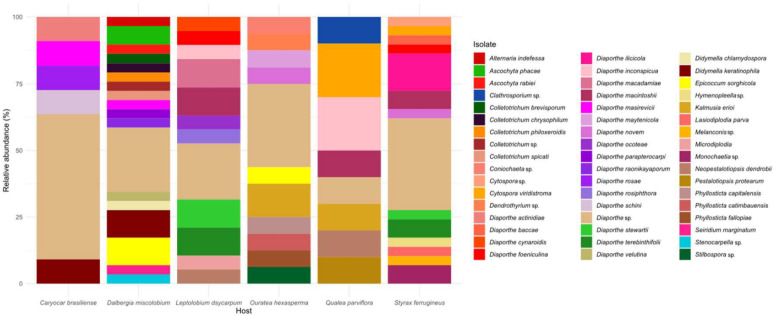
Species-level relative abundance (%) of the cultivable endophytic fungal community of six host species from the Brazilian Cerrado. Figure created with R software version 4.1.1. using the package “*ggplot2*”.

**Figure 6 jof-09-00508-f006:**
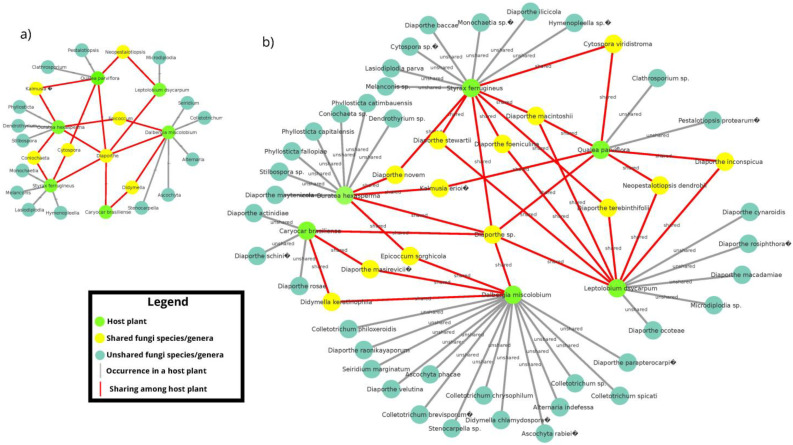
Architecture of the distribution network of endophytic fungi among six host plant species. (**a**) Distribution and sharing of fungi genera among host plant species; (**b**) distribution and sharing of genera and species is demonstrated. Light green nodes represent host plants; dark green nodes show fungal isolates (genus and/or species) not shared among host plants; and yellow nodes represent shared fungal isolates (genus and/or species) among host species. Gray lines indicate the occurrence of single taxa for specific hosts; red lines highlight species sharing. Figure created using the CYTOSCAPE version 3.9.1.

**Figure 7 jof-09-00508-f007:**
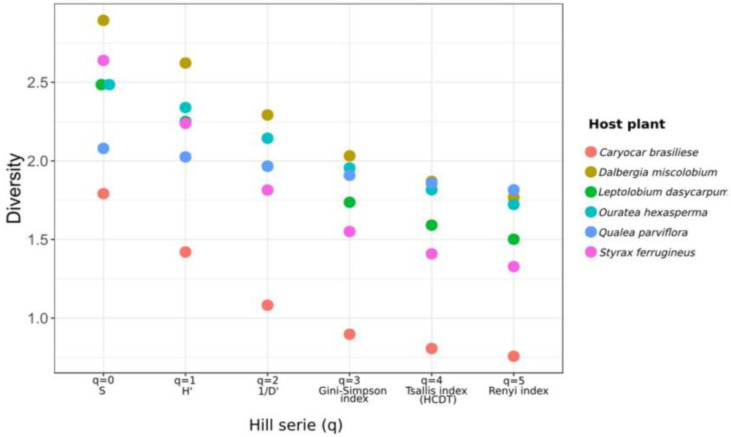
Alpha diversity metrics for the fungal community isolated from six host plant species shown by Hill Series (q). The alpha diversity metric is shown at the X-axis. The Y-axis shows diversity values over the X-axis for each diversity metric employed. q = 0, species richness; q = 1, Shannon–Wiener index (H’); q = 2, Simpson dominance (1/D’); q = 3, Gini–Simpson index; q = 4, Tsallis index (HCDT); q = 5, Renyi index. Figure created with R software version 4.1.1. using the package “*ggplot2*”.

**Figure 8 jof-09-00508-f008:**
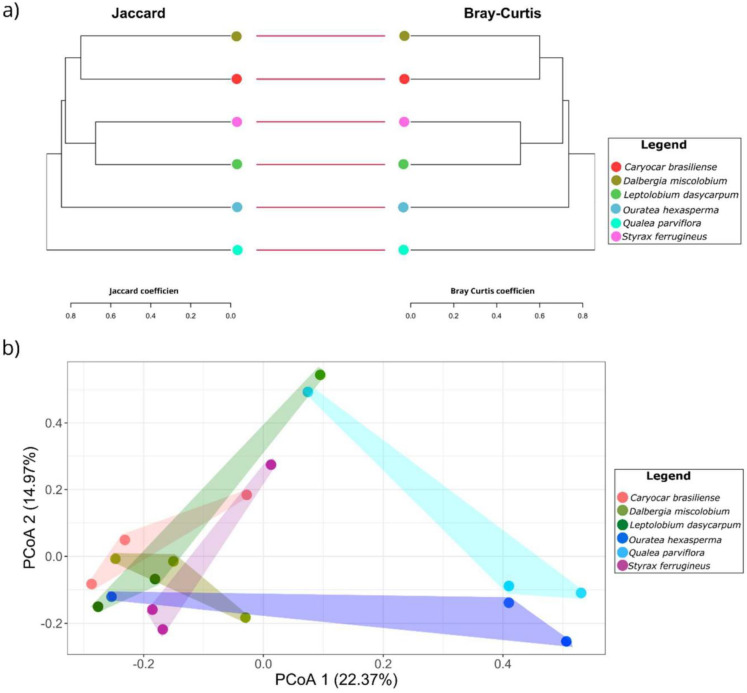
Beta diversity metrics of the endophytic fungi community isolated from six host plant species. In (**a**), dendrograms constructed from the Jaccard distance (binary presence/absence matrix) and Bray–Curtis (matrix of abundance) are shown. The bars between the dendrograms show similar clustering of communities for both metrics. In (**b**), the Principal Coordinate Analysis (PCoA) obtained from the Bray–Curtis distance matrix is shown. The percentage of variation explained by the plotted principal coordinates is indicated on X and Y axes. The colored circles indicate the grouping of the fungal communities of each host plant species. Figure created with R software version 4.1.1. using the package “*ggplot2*”.

**Figure 9 jof-09-00508-f009:**
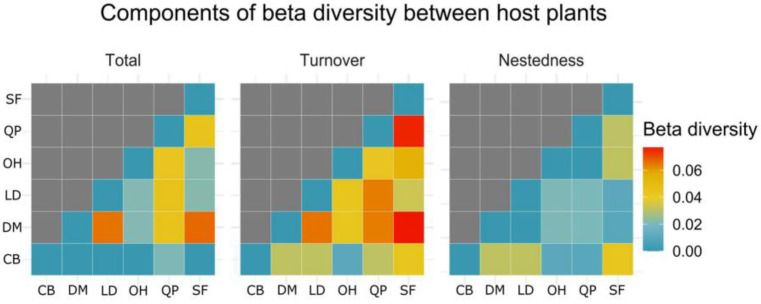
Heatmap showing the pairwise comparisons obtained using the Sorenson index and its components (total, turnover, nestedness) from the endophytic fungi community isolated from six host plant species. The X axis and Y axis show the host species analyzed. The value obtained for each of the components of the Sorensen index is presented on a color scale from blue (=minor) to red (=highest). CB: *Caryocar brasiliense*; DM: *Dalbergia miscolobium*; LD: *Leptolobium dasycarpum*; OH: *Ouratea hexasperma*; QP: *Qualea parviflora*; and SF: *Styrax ferrugineus*. Figure created with R software version 4.1.1. using the package “*ggplot2*”.

**Figure 10 jof-09-00508-f010:**
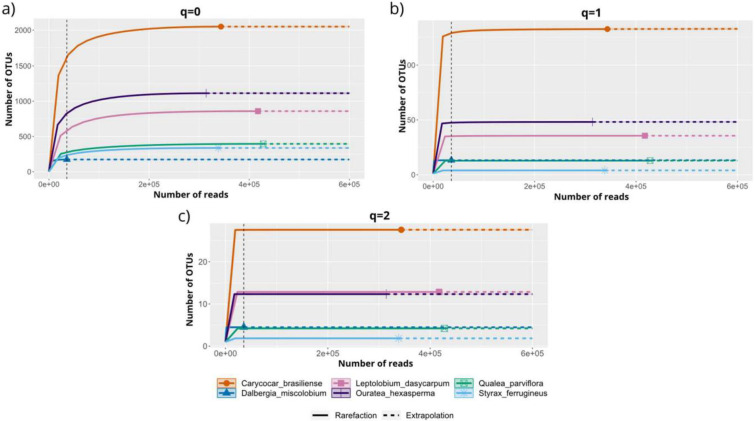
Rarefaction and extrapolation curves based on sample size (number of reads) of the foliar endophytic fungal community associated with six native woody species from the Brazilian Cerrado. (**a**) Species richness (Hill number for q = 0); (**b**) equally abundant species (Hill number for q = 1), and (**c**) dominance (Hill number for q = 2). Diversity curves were constructed using the number of rarefied reads (solid lines) and extrapolated (dashed horizontal lines) with estimates based on sample size. Each curve was extrapolated up to a doubling of its reference sample size. The vertically dashed black line shows the smallest sample size. Figure created with R software version 4.1.1. using the “*iNEXT*” package.

**Figure 11 jof-09-00508-f011:**
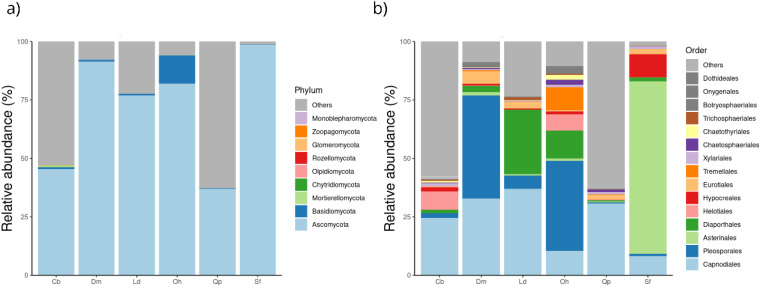
Relative abundance of the leaf endophytic fungal community associated with six Cerrado woody species. In (**a**), the relative abundance is shown at the phylum level, and in (**b**) at the order level. Sequences classified only at the kingdom level and/or in lesser abundance were grouped into “Others”. Cb: *Caryocar brasiliense*; Dm: *Dalbergia miscolobium*; Ld: *Leptolobium dasycarpum*; Oh: *Ouratea hexasperma*; Qp: *Qualea parviflora*; and Sf: *Styrax ferrugineus*.

**Figure 12 jof-09-00508-f012:**
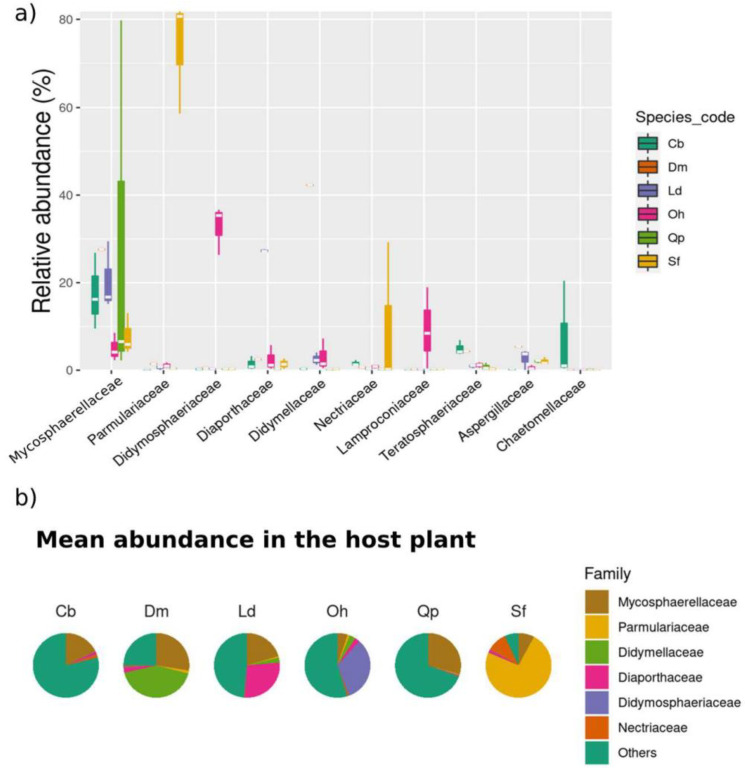
Taxonomic composition of the leaf endophytic fungal community of six host plant species at the family level. In (**a**), a comparison of the most abundant families among the different host plants is shown. In (**b**), the average of the most abundant fungal families in each host species is presented. Cb: *Caryocar brasiliense*; Dm: *Dalbergia miscolobium;* Ld: *Leptolobium dasycarpum*; Oh: *Ouratea hexasperma*; Qp: *Qualea parviflora*; and Sf: *Styrax ferrugineus*.

**Figure 13 jof-09-00508-f013:**
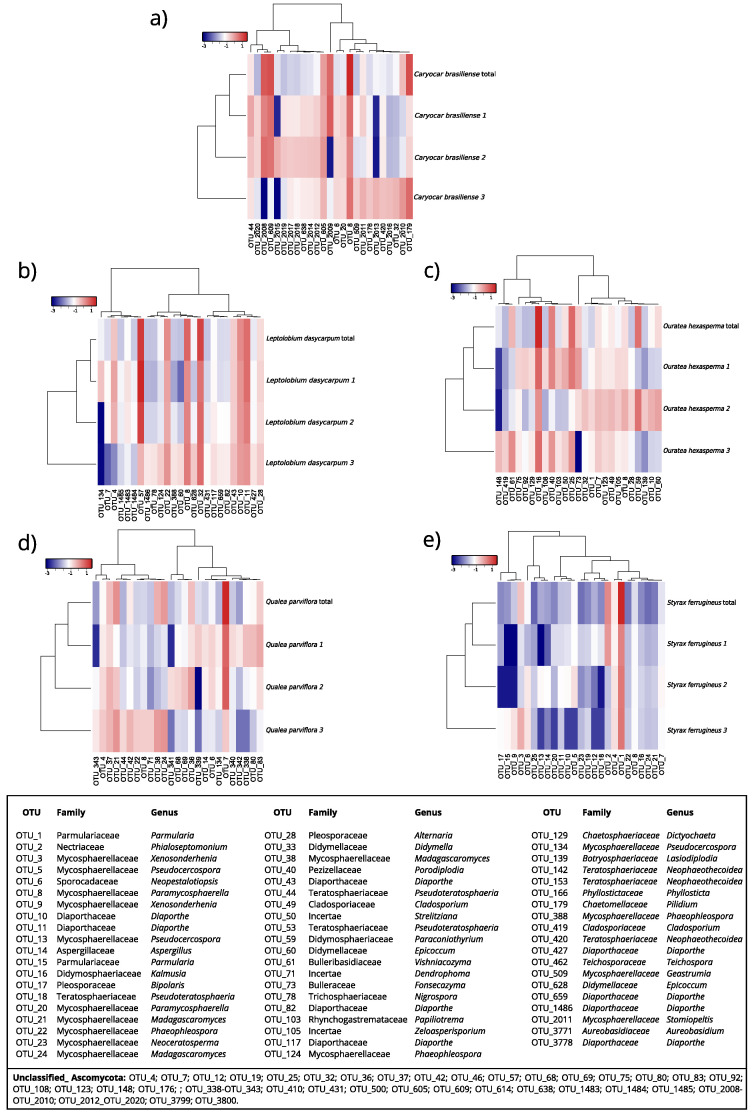
Heatmap representing the relative abundance of the 25 most abundant OTUs for each host plant at the genus level. Each heatmap represents a specific host, as follows: (**a**) *C. brasiliense*, (**b**) *L. dasycarpum*, (**c**) *O. hexasperma*, (**d**) *Q. parviflora*, and (**e**) *S. ferrugineus*. The lines are the sampling effort employed (three replicates per host) and the sum of the sampling effort (total). The columns show the 25 most abundant OTUs. The colors represent the relative abundance of each OTU: the redder the greater the abundance; the bluer the lower the abundance. To facilitate the data visualization, the number of readings, which is equivalent to the abundance, was represented on a logarithmic scale (base 10). Figures were plotted using the “*Heatmap3*” package implemented by R software version 4.1.1.

**Figure 14 jof-09-00508-f014:**
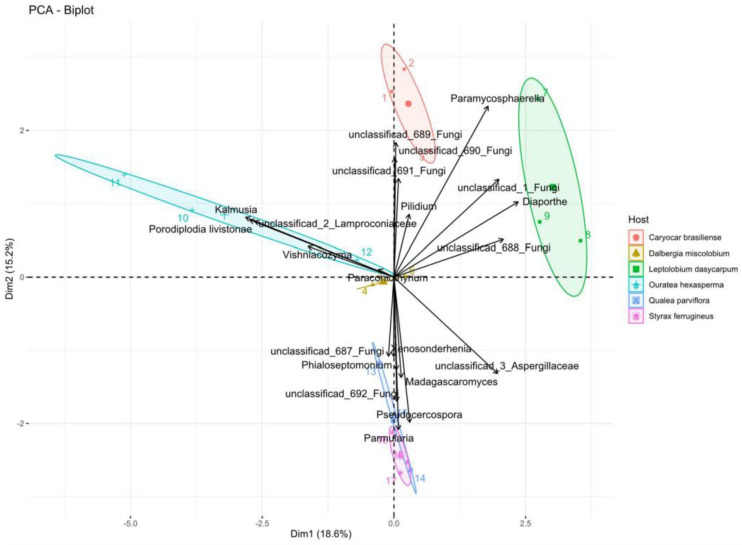
Principal Component Analysis (PCA) of the five most abundant fungal OTUs among the six plant species analyzed. PCA biplot shows the relationship between the abundance (n = reads) of OTUs from fungal endophytes (arrows) and host plants (ellipses). The longer the arrow and the further away from the center, the more influence the abundance of the fungus has. The closer to a host species (cluster), the stronger their relationship. Figure created with R software version 4.1.1. using the package “*ggplot2*”.

**Figure 15 jof-09-00508-f015:**
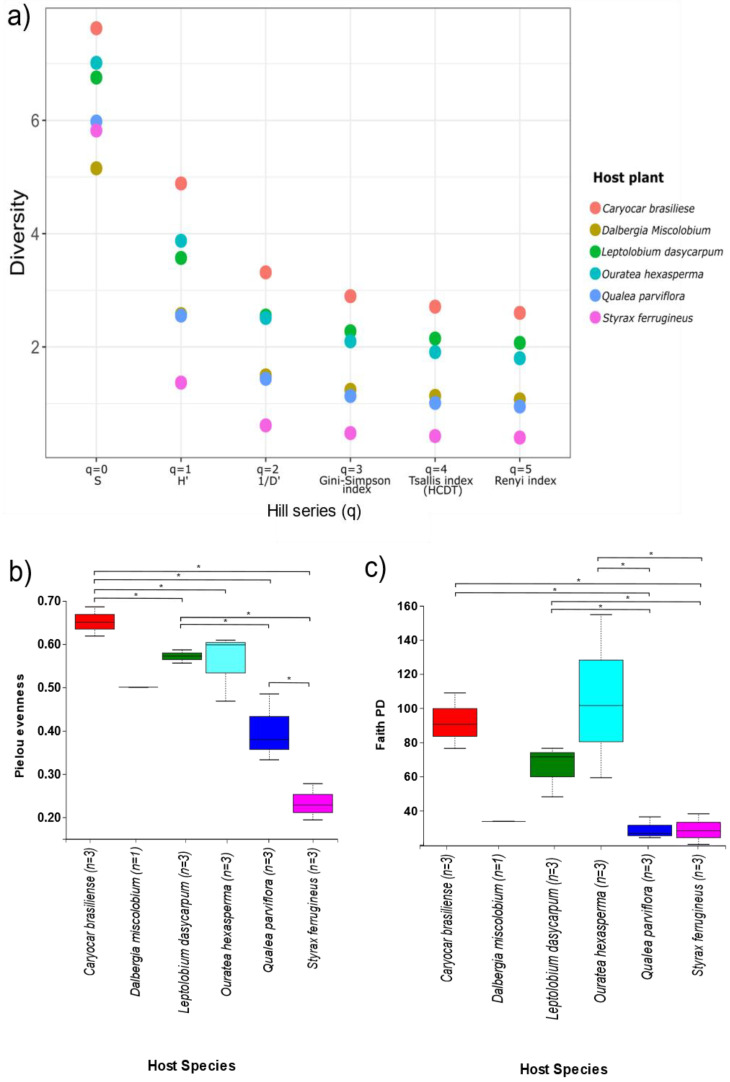
Alpha diversity indices of the community of endophytic fungi associated with six Cerrado native plant species. In (**a**), the Hill series is shown (q = 0, q = 1, q = 2, q = 3, q = 4, q = 5). The alpha diversity metric is displayed at the X axis. The Y axis shows the diversity values obtained over the X axis for each diversity metric employed. q = 0, species richness; q = 1, Shannon–Wiener index (H’); q = 2, Simpson dominance (1/D’); q = 3, Gini–Simpson index; q = 4, Tsallis index (HCDT); q = 5, Renyi index. In (**b**), Pielou’s equity is revealed. In (**c**), Faith’s phylogenetic diversity is demonstrated. The character “*” shows where statistically significant differences were found between the endophytic fungal community and the different host species (Kruskal–Wallis paired test, *p* < 0.05). The boxplots represent the inter-hip range (IQR) between the first and third quartiles (25th and 75th percentiles, respectively). The tails indicate the lowest and highest values found, and the middle horizontal lines inside the boxes are means. Figure created with R software version 4.1.1. using the “*ggplot2*” package.

**Figure 16 jof-09-00508-f016:**
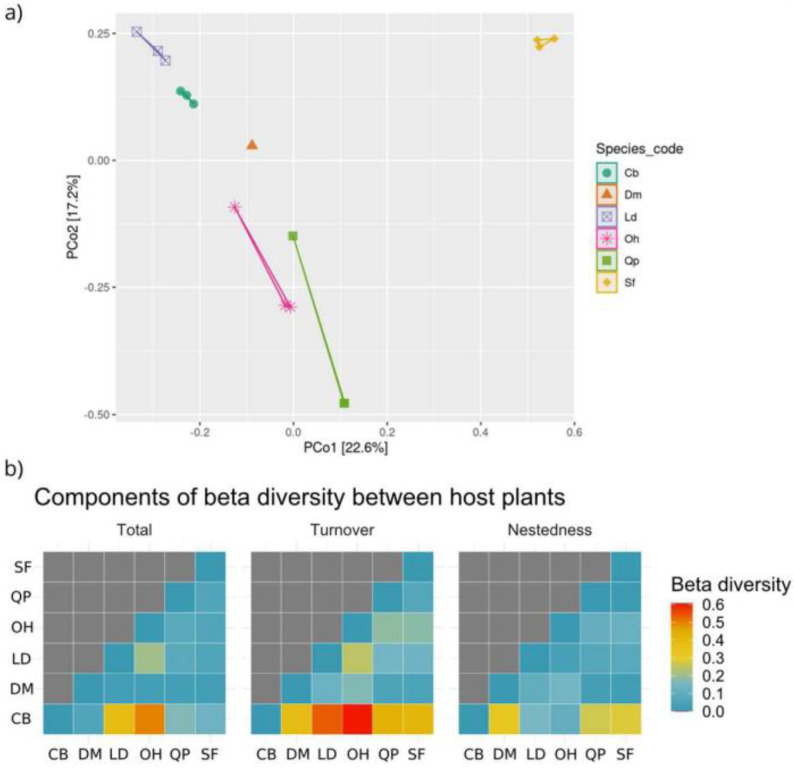
Metrics of beta diversity of the endophytic fungal community associated with six Cerrado woody plant species. In (**a**), Principal Coordinate Analysis (PCoA) derived from the Bray–Curtis distance is shown. The percentage of variation explained by the plotted principal coordinates is indicated at the X and Y axes. The colored symbols indicate the grouping of the fungal communities from each host plant species. In (**b**), the heatmap showing the paired comparisons obtained from the Sorenson index and its components (total, turnover, nestedness) of the community of endophytic fungi isolated from six plant species are shown. The X axis and the Y axis show the analyzed host species. The values obtained in each of the components of the Sorensen index are presented in a color scale: blue = minor; red = highest. CB: *Caryocar brasiliense*; DM: *Dalbergia miscolobium*; LD: *Leptolobium dasycarpum*; OH: *Ouratea hexasperma*; QP: *Qualea parviflora*, and SFf: *Styrax ferrugineus*. Figure created with R software version 4.1.1. using the “*ggplot2*” package.

**Figure 17 jof-09-00508-f017:**
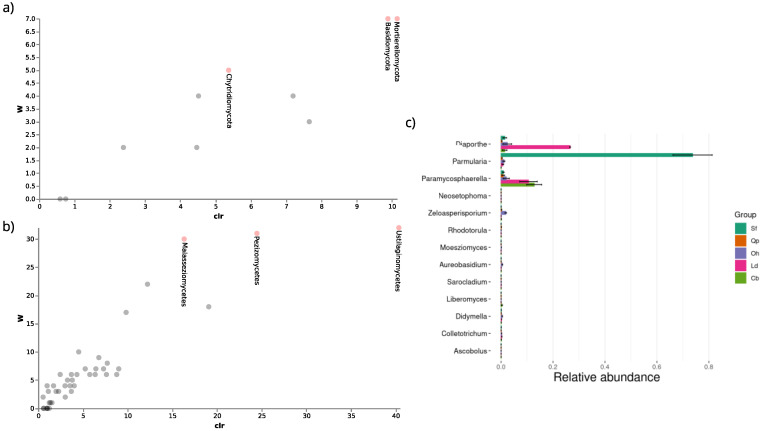
Differential analysis of abundance at different taxonomic levels of the endophytic fungal communities from six hosts. ANCOM generated volcanic plots showing differentially abundant features among the hosts *C. brasiliense*, *D. miscolobium*, *L. dasycarpum*, *Q. parviflora*, *O. hexasperna*, and *S. ferrugineus*. The “W” is the ANCOM test statistic, which demonstrates the number of times the null hypothesis is rejected by the analysis. The higher the W value, the greater the probability of a characteristic differing statistically. The “clr” shows the change in size between groups compared by the test. The statistically significant characteristics found among the host species are labeled in (**a**) phylum and (**b**) subphylum/class level. In (**c**), taxa at the genus level differentially abundant among the host species are shown by Metastat. At the Y axis, the main differentially abundant genera are shown, the relative abundances of which are displayed at the X axis. The host plant *D. miscolobium* was not considered, as Metastat requires at least two biological replicates for each group. Cb: *C. brasiliense*; Ld: *L. dasyrcarpum*; Oh: *O. hexasperma*; Qp: *Q. parviflora*, and Sf: *S. ferrugineus*.

**Figure 18 jof-09-00508-f018:**
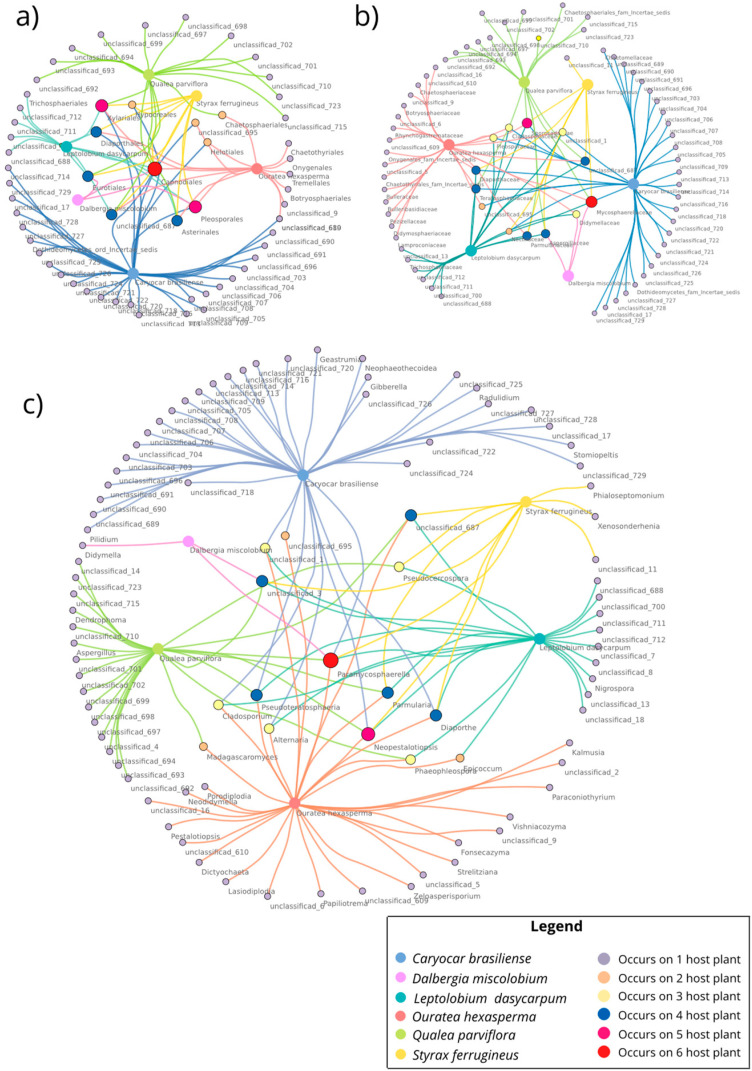
Co-occurrence network at different taxonomic levels of the leaf endophytic fungal community of six host plant species. The nodes represent the fungal OTUs at (**a**) order, (**b**) family, and (**c**) genus level. The color of nodes indicates the occurrence/co-occurrence among host plants. The edges indicate each host plant where OTUs occur/co-occur. To facilitate the data visualization, the abundance of OTUs was not considered. Figure created using CYTOSCAPE version 3.9.1.

**Table 1 jof-09-00508-t001:** Foliar nutrient contents and specific leaf area (SLA–cm^2^/g) of *Caryocar brasiliense*, *Dalbergia miscolobium*, *Leptolobium dasycarpum*, *Qualea parviflora*, *Ouratea hexasperma*, and *Styrax ferrugineus*. Mean values ± standard deviation. Different letters represent statistical differences (*p* < 0.05) set by One-way Anova.

Species	N (dag/Kg)	P (dag/Kg)	K (dag/Kg)	Ca (dag/Kg)	Mg (dag/Kg)	S (dag/Kg)	Al (dag/Kg)	SLA (cm^2^/g)
*Caryocar brasiliense*	1.25 ± 0.06 ^b^	0.06 ± 0.00 ^a^	0.30 ± 0.04 ^ab^	0.36 ± 0.08 ^c^	0.18 ± 0.04 ^bd^	0.08 ± 0.00 ^ab^	280.34 ± 44.11 ^a^	66.49 ± 5.84 ^bc^
*Dalbergia miscolobium*	1.95 ± 0.09 ^a^	0.06 ± 0.00 ^a^	0.29 ± 0.02 ^a^	0.18 ± 0.03 ^a^	0.14 ± 0.02 ^ab^	0.08 ± 0.00 ^ab^	113.00 ± 27.15 ^a^	94.04 ± 15.43 ^a^
*Leptolobium dasycarpum*	1.78 ± 0.14 ^a^	0.06 ± 0.00 ^a^	0.32 ± 0.05 ^ab^	0.17 ± 0.04 ^a^	0.08 ± 0.03 ^a^	0.07 ± 0.01 ^a^	172.66 ± 41.66 ^a^	63.26 ± 12.48 ^bc^
*Qualea parviflora*	1.27 ± 0.10 ^b^	0.06 ± 0.00 ^ab^	0.34 ± 0.05 ^ab^	0.32 ± 0.02 ^ac^	0.17 ± 0.02 ^bd^	0.08 ± 0.01 ^ab^	12,345.41 ± 1001.30 ^b^	85.06 ± 14.09 ^ac^
*Ouratea hexasperma*	1.17 ± 0.15 ^b^	0.05 ± 0.00 ^b^	0.38 ± 0.06 ^b^	0.66 ± 0.14 ^b^	0.25 ± 0.03 ^c^	0.08 ± 0.01 ^ab^	106.08 ± 27.07 ^a^	56.34 ± 10.52 ^b^
*Styrax ferrugineus*	1.33 ± 0.09 ^b^	0.06 ± 0.00 ^a^	0.27 ± 0.02 ^a^	0.40 ± 0.12 ^c^	0.21 ± 0.06 ^cd^	0.09 ± 0.00 ^b^	242.27 ± 61.44 ^a^	63.16 ± 10.26 ^bc^

**Table 2 jof-09-00508-t002:** Sequence identity (%) of endophytic leaf fungi from Cerrado woody species compared with other sequences from the GenBank using BLAST analysis. Isolates with an identity percentage below 97% were classified only at the genus level.

Host Species	Isolate ID	Closest GenBank Match	Match	Identity (%)	Query Coverage (%)
*Caryocar brasiliense*	CB1629_2 CB1629_22 CB1629_25 CB1629_3 CB1629_31 CB995_15 CB995_2 CB995_22 CB995_3 CB995_6 CB995_7	*Diaporthe* sp. *Diaporthe schini* *Diaporthe* sp. *Diaporthe* sp. *Didymella keratinophila* *Diaporthe rosae* *Diaporthe* sp. *Diaporthe* sp. *Diaporthe* sp. *Diaporthe actinidiae* *Diaporthe masirevicii*	NR_111849.1 NR_111861.1 NR_147537.1 NR_145303.1 NR_158275.1 MG828894.1 EU552122.1 MH171064.1 NR_137105.1 KC145886.1 NR_147534.1	95% 99% 95% 94% 99% 99% 95% 96% 92% 99% 98%	88% 100% 86% 95% 100% 100% 100% 99% 100% 100% 100%
*Dalbergia miscolobium*	DM1044_2 DM1044_23 DM1044_3 DM1044_6 DM1044_16 DM1044_8 DM1044_19 DM1044_9 DM1044_5 DM18_1 DM18_12 DM18_2 DM18_8 DM2226_1 DM2226_10 DM2226_11 DM2226_11_1 DM2226_13 DM2226_14 DM2226_15 DM2226_19 DM2226_22 DM2226_29 DM2226_3 DM2226_4 DM2226_17 DM2226_5 DM2226_6 DM18_6	*Colletotrichum spicati* *Didymella chlamydospora* *Diaporthe* sp. *Alternaria indefessa* *Diaporthe masirevicii* *Didymella keratinophila* *Diaporthe* sp. *Diaporthe raonikayaporum* *Didymella keratinophila* *Epicoccum sorghicola* *Diaporthe* sp. *Epicoccum sorghicola* *Diaporthe* sp. *Diaporthe* sp. *Diaporthe parapterocarpi* *Ascochyta phacae* *Diaporthe velutina* *Colletotrichum* sp. *Seiridium marginatum* *Colletotrichum chrysophilum* *Colletotrichum philoxeroidis* *Stenocarpella* sp. *Diaporthe* sp. *Ascochyta rabiei* *Diaporthe* sp. *Ascochyta phacae* *Colletotrichum brevisporum* *Epicoccum sorghicola* *Didymella keratinophila*	OL842171.1 MK836111.1 MH855768.1 MH861641.1 KJ197276.1 NR_158275.1 EU552122.1 NR_111860.1 NR_158275.1 OK442368.1 MH864501.1 OK442368.1 MN152927.1 MH855768.1 NR_168152.1 MH857437.1 NR_152470.1 MK541032.1 KT949914.1 NR_160821.1 OL842188.1 NR_173403.1 NR_168240.1 EU167600.1 EU552122.1 MH857437.1 KC790943.1 OK442368.1 NR_158275.1	99% 98% 95% 99% 98% 99% 93% 99% 99% 98% 93% 98% 93% 95% 99% 98% 97% 92% 98% 99% 97% 93% 94% 99% 90% 98% 98% 99% 99%	96% 100% 100% 100% 100% 100% 100% 100% 100% 100% 100% 100% 99% 100% 100% 99% 100% 100% 99% 100% 96% 100% 92% 99% 100% 99% 95% 100% 100%
*Leptolobium dsycarpum*	LD2293_12 LD2293_13 LD2293_23 LD2293_3 LD2293_4 LD2293_7 LD2299_1 LD2299_2 LD2299_3 LD2299_35 LD2299_4 LD2299_5 LD2299_6 LD2304_1 LD2304_2 LD2304_30 LD2304_34 LD2304_35 LD2304_50	*Diaporthe stewartii* *Diaporthe stewartii* *Diaporthe cynaroidis* *Diaporthe terebinthifolii* *Diaporthe* sp. *Microdiplodia* sp. *Diaporthe macadamiae* *Diaporthe terebinthifolii* *Diaporthe ocoteae* *Diaporthe macadamiae* *Diaporthe* sp. *Diaporthe rosiphthora* *Diaporthe* sp. *Diaporthe* sp. *Diaporthe foeniculina* *Neopestalotiopsis dendrobii* *Diaporthe macintoshii* *Diaporthe macintoshii* *Diaporthe inconspicua*	MH855768.1 MH855768.1 EU552122.1 NR_111862.1 NR_147596.1 DQ885897.1 NR_168240.1 NR_111862.1 NR_147596.1 NR_168240.1 NR_111849.1 MT311197.1 MH171064.1 NR_168240.1 NR_145303.1 MK993572.1 NR_147539.1 NR_147539.1 NR_111849.1	99% 96% 96% 99% 93% 96% 98% 99% 98% 98% 90% 99% 96% 95% 97% 99% 98% 98% 99%	100% 100% 100% 99% 100% 100% 98% 100% 100% 100% 99% 100% 100% 96% 100% 100% 99% 100% 99%
*Ouratea hexasperma*	OH1078_22 OH1078_12 OH1078_2 OH1078_3 OH1078_31 OH1078_24 OH2206_31 OH2208_3 OH2208_4 OH2262_2 OH2262_30 OH2262_31 OH2262_4 OH2262_40 OH2262_5 OH2262_6	*Kalmusia erioi* *Phyllosticta catimbauensis* *Diaporthe novem* *Diaporthe* sp. *Kalmusia erioi* *Dendrothyrium* sp. *Phyllosticta fallopiae* *Stilbospora* sp. *Phyllosticta capitalensis* *Diaporthe maytenicola* *Diaporthe* sp. *Epicoccum sorghicola* *Coniochaeta* sp. *Diaporthe* sp. *Diaporthe* sp. *Diaporthe* sp.	MN473058.1 NR_156631.1 MH864503.1 NR_120138.1 MN473058.1 JX496097.1 AB454307.1 KF570166.1 OL957169.1 NR_137826.1 EU552122.1 OK442368.1 NR_173009.1 NR_147596.1 NR_158416.1 MH855768.1	99% 99% 98% 93% 97% 93% 100% 88% 100% 97% 92% 98% 90% 93% 93% 95%	100% 88% 100% 97% 97% 98% 99% 100% 100% 100% 100% 100% 99% 100% 100% 100%
*Qualea parviflora*	QP1131_11 QP1646_1 QP1646_11 QP1646_13 QP1646_3 QP1646_4 QP65_0 QP65_30 QP65_4 QP971_2	*Pestalotiopsis protearum* *Diaporthe macintoshii* *Neopestalotiopsis dendrobii* *Diaporthe* sp. *Diaporthe inconspicua* *Diaporthe inconspicua* *Kalmusia erioi* *Cytospora viridistroma* *Cytospora viridistroma* *Clathrosporium* sp.	JN712498.1 NR_147539.1 MK993572.1 NR_147539.1 NR_111849.1 NR_111849.1 MN473058.1 MN172408.1 MN172408.1 NR_153908.1	99% 98% 99% 96% 99% 99% 97% 99% 99% 88%	100% 100% 100% 100% 100% 100% 97% 100% 100% 86%
*Styrax ferrugineus*	SF1034_3 SF1034_4 SF134_2 SF134_22 SF134_25 SF134_3 SF134_3_1 SF134_4 SF134_6 SF134_7 SF1697_10 SF1697_16 SF1697_17 SF1697_3 SF1697_13 SF1697_7 SF1697_8_1 SF1697_8 SF1697_15 SF1756_12 SF1756_13_1 SF1756_18 SF1756_2 SF1756_3 SF1756_4 SF1756_5 SF1756_6 SF1756_8 SF1756_9	*Cytospora viridistroma* *Diaporthe* sp. *Diaporthe macintoshii* *Diaporthe macintoshii* *Diaporthe foeniculina* *Hymenopleella* sp. *Diaporthe* sp. *Diaporthe ilicicola* *Monochaetia* sp. *Monochaetia* sp. *Diaporthe novem* *Diaporthe* sp. *Diaporthe stewartii* *Lasiodiplodia parva* *Cytospora* sp. *Melanconis* sp. *Diaporthe* sp. *Diaporthe terebinthifolii* *Diaporthe* sp. *Diaporthe baccae* *Diaporthe ilicicola* *Diaporthe* sp. *Diaporthe* sp. *Diaporthe ilicicola* *Diaporthe* sp. *Diaporthe* sp. *Diaporthe ilicicola* *Diaporthe* sp. *Diaporthe terebinthifolii*	MN172408.1 NR_111857.1 NR_147539.1 NR_147539.1 NR_145303.1 KT949901.1 NR_137825.1 MH171064.1 LC146750.1 LC146750.1 MH864503.1 EU552122.1 MH855768.1 MH861166.1 MK912135.1 MN784964.1 MH855768.1 NR_111862.1 MH855768.1 NR_152458.1 MH171064.1 NR_172435.1 MH171064.1 MH171064.1 MH171064.1 NR_137126.1 MH171064.1 NR_111849.1 NR_111862.1	99% 93% 98% 98% 97% 90% 96% 98% 95% 95% 99% 90% 98% 99% 98% 90% 95% 99% 96% 97% 97% 96% 95% 97% 95% 96% 97% 95% 99%	100% 100% 100% 100% 100% 100% 100% 99% 90% 90% 100% 100% 100% 100% 100% 100% 100% 100% 100% 100% 92% 100% 89% 99% 100% 100% 99% 90% 100%

**Table 3 jof-09-00508-t003:** Number of Operational Taxonomic Units (OTUs) and taxonomic attributions obtained by metabarcoding analysis of the foliar endophytic fungal community of six host plant species.

Plant Host	Read	OTUs	Phyla	Classes	Orders	Families	Genera
*Caryocar brasiliense*	343,535	2050	8	27	64	162	309
*Dalbergia miscolobium*	35,782	173	3	13	30	52	68
*Leptolobium dasycarpum*	417,491	859	7	25	64	131	169
*Ouratea hexasperma*	314,104	1113	9	22	58	142	243
*Qualea parviflora*	427,863	395	5	11	32	75	116
*Styrax ferrugineus*	338,354	337	4	14	42	87	114
Total	1,877,129	3821	9	35	86	214	435

## Data Availability

The nucleotide sequences generated from the fungi isolated in this study were deposited in GenBank under the accession numbers OP922124–OP922237.

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
