# Peer review of "How Deep Can the Endophytic Mycobiome Go? A Case Study on Six Woody Species from the Brazilian Cerrado"

_jof, 2023, doi:10.3390/jof9050508_

Round 1

Reviewer 1 Report

I have suggested some avenues they should follow to improve mostly the discussion

Author Response

Response to the reviewers’ point-by-point

Reviewer’s comment:

Please add this reference as highly relevant:

Bhunjun, C.S., Phukhamsakda, C., Hyde, K.D. et al. (2023) Do all fungi have ancestors with endophytic lifestyles? Fungal Diversity

Response: Effected (please, see line 48).

  1. Bhunjun C.S., Phukhamsakda C., Hyde K.D. et al.Do all fungi have ancestors with endophytic lifestyles?. Fungal Divers.2023. https://doi.org/10.1007/s13225-023-00516-5.

Reviewer’s comment:

Please add this reference:

Tao, G.; Liu, Z. Y.; Hyde, K. D.; Liu, X. Z.; Yu, Z. N. (2008) Whole rDNA analysis reveals novel and endophytic fungi in Bletilla ochracea (Orchidaceae). Fungal Diversity 33, 101–122.

Response: Effected (please, see line 49).

  1. Tao G., Liu, Z.Y., Hyde K.D., Liu X.Z., Yu Z.N. Whole rDNA analysis reveals novel and endophytic fungi in Bletilla ochracea (Orchidaceae). Fungal Divers. 2008; 33, 101–122.

Reviewer’s comment:

add Bhunjun et al. 2023

Response: Effected (please, see line 51).

  1. Bhunjun C.S., Phukhamsakda C., Hyde K.D. et al.Do all fungi have ancestors with endophytic lifestyles? Fungal Divers.2023. https://doi.org/10.1007/s13225-023-00516-5.

Reviewer’s comment:

To add

In many cases endophytes produce appressoria which allow them to infect their respective hosts (Chathana et al. 2021a, b).

Chethana KWT, Jayawardena RS, Chen YJ, Konta S et al. (2021) Appressorial interactions with host and their evolution. Fungal Diversity 110: 75–107

Response: The information has been included (please, see lines 51-52), and the references (Chethana et al. 2021; Bhunjun et al. 2023) provided.

“In many cases, endophytic fungi depend on host plants to complete part or all of their life cycle [16-18], and produce appressoria to allow infection of the respective hosts [11, 19].”

  1. Bhunjun C.S., Phukhamsakda C., Hyde K.D. et al.Do all fungi have ancestors with endophytic lifestyles?. Fungal Divers.2023. https://doi.org/10.1007/s13225-023-00516-5.

  1. Chethana K.W.T., Jayawardena R.S., Chen Y.J., Konta S., Tibpromma S., Phukhamsakda C., Abeywickrama P.D., Samarakoon M.C., Senwanna C., Mapook A., Tang X., Gomdola D., Marasinghe D.S., Padaruth O.D., Balasuriya A., Xu J., Lumyong S., Hyde K.D. Appressorial interactions with host and their evolution. Fungal Divers. 2021; 110:1–33.

Reviewer’s comment:

add here

and the fact that most endophyte studies only use ITS which is inadequate (Ko et al. 2011)

Ko TWK, Stephenson SL, Bahkali AH, Hyde KD (2011) From morphology to molecular biology: can we use sequence data to identify fungal endophytes? Fungal Diversity 50:113-120.

Response: Effected (please, see lines 73-74).

  1. Ko T.W.K, Stephenson S.L., Bahkali A.H., Hyde K.D. From morphology to molecular biology: can we use sequence data to identify fungal endophytes? Fungal Divers. 2011; 50:113-120.

Reviewer’s comment:

add Dissanayake et al. 2018

Dissanayake AJ, Purahong W, Wubet T, Hyde KD, Zhang W, Xu HY, Zhang GJ, Fu CY, Liu M, Xing QK, Li XH, Yan JY. (2018) Direct comparison of culture-dependent and culture-independent molecular approaches reveal the diversity of fungal endophytic communities in stems of grapevine (Vitis vinifera). Fungal Divers 90: 85-107.

Response: In line with the reviewer’s comments, a recent publication by Durán et al. (2021) has been provided (please, see lines 75 and 77).

  1. Durán M., San Emeterio L., Canals R.M. Comparison of Culturing and Metabarcoding Methods to Describe the Fungal Endophytic Assemblage of Brachypodium rupestre Growing in a Range of Anthropized Disturbance Regimes. Biology (Basel); 2021; 10(12):1246. doi: 10.3390/biology10121246.

Reviewer’s comment:

of

Response: Thank you. Corrected (please, see line 133).

Reviewer’s comment:

thee are much higher

Response: The information displayed in Table 3 was double checked.

Reviewer’s comment:

add Ko et al 2011 as above

Response: Effected (please, see line 678).

  1. Ko T.W.K, Stephenson S.L., Bahkali A.H., Hyde K.D. From morphology to molecular biology: can we use sequence data to identify fungal endophytes? Fungal Divers. 2011; 50:113-120.

Reviewer’s comment:

Most of these genera are quick growing species that would outcompete the slow growing species

(ref)

Response: The information has been provided (please, see lines 692-695).

“Other genera found in our study (Alternaria, Phyllosticta, Epicoccum, Colletotrichum, Fusarium, Stenocarpella, and Lasiodiplodia) considered fast-growing [11] have already been reported as endophytic fungi that occur in Cerrado species [31, 64-65], in addition to other ecosystems [14, 67].”

Reviewer’s comment:

I did not mean to delete this, but wanted to point something out.

You can also compare this with Parungao et al. (2002). She looked at the saprobes on several different species of leaves in a tropical forest and found similar results to you, but the species were very diferent.

Parungao MM, Fryar SC, Hyde KD (2002) Diversity of fungi on rainforest litter in North Queensland, Australia. Biodiversity & Conservation volume 11, 1185–1194

You should also compare with Paulus et al. (2006) who use particle filtration to show a high diversity.

Paulus BC, Kanowski J, Gadek PA, Hyde KD (2006) Diversity and distribution of saprobic microfungi in leaf litter of an Australian tropical rainforest. Mycol Res 110:1441–1454.

Response: In line with the reviewer’s comments, the sentence has been rephrased and Parungao et al. (2002) included (please, see lines 701-711).

“Most isolated fungal genera/species were found in only one host plant species. This is highlighted by the genera/species sharing network between host plants (Figure 6). Variability of fungal species isolated from different host plant species in the same habitat was reported in previous studies of cultivable endophytic fungi [23, 31, 74-75], and leaf litter fungi [76]. One explanation for these variations refers to the “environmental filtering” effect exerted by the host plant identity in conjunction with the plant-fungus symbiont coevolution relationship [20-21, 77]. In this context, the phytochemistry and nutritional resources from plant tissues may act as selection factors for endophytic plant colonization [20-21, 77]. Moreover, genotype and phenotype of host plants may drastically influence the occurrence, diversity, and abundance of species within the endophytic fungi mycobiome. Although our sampling effort was not sufficient to cover the full diversity of cultivable endophytic fungi, a larger sample size could have resulted in fewer fungi appearing to be host-specific as reported by [76].”

  1. Parungao M.M., Fryar S.C. & Hyde K.D. Diversity of fungi on rainforest litter in North Queensland, Australia. Biodivers Conserv. 2002; 11, 1185–1194. https://doi.org/10.1023/A:1016089220042.

Reviewer’s comment:

add Jayawardena RS, Purahong W, Zhang W, Wubet T, Li XH, Liu M, Zhao WS, Hyde KD, Liu JH, Yan JY. (2018) Biodiversity of fungi on Vitis vinifera L. revealed by traditional and high-resolution culture-independent approaches. Fungal Divers. 2018; 90:1–84.

Response: Effected (please, see line 721).

  1. Jayawardena R.S., Purahong W., Zhang W., Wubet T., Li X.H., Liu M., Zhao W.S., Hyde K.D., Liu J.H., Yan J.Y. Biodiversity of fungi on Vitis vinifera L. revealed by traditional and high-resolution culture-independent approaches. Fungal Divers. 2018; 90:1–84.

Reviewer’s comment:

please give some reasons for these results

Resposta: This finding was reported by Daghino et al. (2022). Possible reasons for this has not been provided to avoid excessive speculation.

  1. Daghino S., Martino E., Voyron S., Perotto S. Metabarcoding of fungal assemblages in Vaccinium myrtillus endosphere suggests colonization of above-ground organs by some ericoid mycorrhizal and DSE fungi. Sci Rep. 2022; 12(1): 11013. doi: 10.1038/s41598-022-15154-1.

Reviewer’s comment:

add a paragraph on the reasons for the difference and if the HTS really shows species that are endphytes

Resposta: A paragraph containing these information has been included (please, see lines 795-798).

“The lack of shared groups highlights the importance of the host identity factor in the composition of endophytic fungal communities among Cerrado plants, since host species may be exerting an environmental filtering effect [20-21, 77]. In this sense, the genotype and phenotype of the host plant may be selecting the pool of species that will colonize it endophytically.”

Reviewer’s comment:

Format

Response: Effected.

Lines 898-899

  1. Zuo Y., Li X., Yang J., Liu J., Zhao L., He X. Fungal endophytic community and diversity associated with desert shrubs driven by plant identity and organ differentiation in extremely arid desert ecosystem. J Fungi. 2021; 7(7): 578.

Lines 1046-1047

  1. Yang J.H., Oh S.Y., Kim W., Hur J.S. Endolichenic fungal community analysis by pure culture isolation and metabarcoding: a case study of Parmotrema tinctorum. Mycobiol. 2022; 50(1): 55-65. doi: 10.1080/12298093.2022.2040112.

Lines 1081-1083

  1. Zhang H., Wei T.P., Li L.Z., Luo M.Y., Jia W.Y., Zeng Y., Jiang Y.L., Tao G.C. Multigene phylogeny, diversity and antimicrobial potential of endophytic sordariomycetes from Rosa roxburghii. Front Microbiol. 2021; 12: 755919. doi: 10.3389/fmicb.2021.755919.

Reviewer 2 Report

The paper "How deep can endophytic mycobiome go? a case study on six woody species from the Brazilian Cerrado" by Jefferson Brendon Almeida dos Reis et al. describe the diversity of fungi associated with six woody species including Caryocar brasiliense, Dalbergia miscolobium, Leptolobium dasycarpum, Qualea parviflora, Ouratea hexasperma, and Styrax ferrugineus using both culture-dependent and culture-independent approaches. 

The content of the manuscript is clear and well written.

I would recommend improving the accuracy of species identification by the addition of one or more genetic markers and construct the phylogenetic tree.

Detailed comments are also included in the manuscript file attached. 

Author Response

Response to the reviewers’ point-by-point

Reviewer’s comment:

The ICTF recommends that scientific names at all taxonomic ranks should be italicized (doi.org/10.1186/s43008-020-00048-6). Please correct all text.

Response: Thank you for the observation. Corrected throughout the manuscript.

Reviewer’s comment:

In my opinion this is not necessary.

Response: As far as we are concerned, this is an important information because it addresses how we recovered endophytic fungi from plant tissue and how the cultures were purified (please, see line 134).

Reviewer’s comment:

I would recommend improving the accuracy of species identification by the addition of one or more genetic markers.

Response: Thank you for the advice. As the ITS region is the most commonly used genetic marker for molecular identification of fungi in environmental sequencing and molecular ecology studies, the ITS region was chosen to reveal the diversity of foliar endophytic fungi from six woody species of the Brazilian Cerrado considering a large-scale taxonomic identification. However, ITS limitations have been provided in the manuscript in lines 678-685. Additional genetic markers will be considered in future studies as recommended by the reviewer.

Reviewer’s comment:

number

Response: Corrected (please, see line 184).

Reviewer’s comment:

Add space

Response: Effected (please, see lines 221-222).

Reviewer’s comment:

please see my comment above (Lines 149, 150)

Response: Thank you again for the advice. Additional genetic markers will be considered in future studies as recommended by the reviewer.

Reviewer’s comment:

Add space

Response: Effected (please, see line 817).

Reviewer’s comment:

number

Response: Corrected (please, see line 824).

Reviewer’s comment:

Please check the reference style according to the format for JoF.

Response: The reference style was double checked according to the JoF guideline.